# On the Convergence Rate of LoRA Gradient Descent

**Siqiao Mu** [1]   **Diego Klabjan** [2]

## Abstract

The low-rank adaptation (LoRA) algorithm for fine-tuning large models has grown popular in recent years due to its remarkable performance and low computational requirements. LoRA trains two "adapter" matrices that form a low-rank representation of the model parameters, thereby massively reducing the number of parameters that need to be updated at every step. Although LoRA is simple, its convergence is poorly understood due to the lack of Lipschitz smoothness, a key condition for classic convergence analyses. As a result, current theoretical results only consider asymptotic behavior or assume strong boundedness conditions which artificially enforce Lipschitz smoothness. In this work, we provide for the first time a non-asymptotic convergence analysis of the *original LoRA gradient descent* algorithm, which reflects widespread practice, without such assumptions. Our work relies on three key steps: i) reformulating the problem in terms of the outer product of the stacked adapter matrices, ii) a modified descent lemma for the "Lipschitz-like" reparametrized function, and iii) controlling the learning rate. With this approach, we prove that the minimum gradient norm under LoRA gradient descent converges at rate $O(\frac{1}{\log T})$, where $T$ is the number of iterations.

## 1. Introduction

Modern applications of large language models (LLMs) typically involve self-supervised pretraining of a large foundation model, which is later fine-tuned on a smaller task-specific dataset through supervised learning. Parameter efficient fine-tuning methods, which only train a small num-

ber of parameters, can efficiently adapt LLMs to a multitude of downstream tasks while maintaining comparable performance to fine-tuning of the full model parameters. One highly popular method is LoRA, or Low-Rank Adaptation (Hu et al., 2022), which trains a low-rank representation of the model parameters. Specifically, once the pretraining stage yields a model weight matrix $W_0$, LoRA only updates low-rank *adapter* matrices $A$, $B$, such that the final weights are $W_0 + BA$. The matrices $A$ and $B$ can be updated with any optimization method, including gradient descent. However, even this simple algorithm is challenging to analyze; as observed in (Ghiasvand et al., 2025; Malinovsky et al., 2024), even if the original loss function is Lipschitz smooth, the *new* loss function— reparametrized in terms of $A$ and $B$— is not necessarily Lipschitz smooth, a major obstacle to applying classic convergence analysis techniques to LoRA.

As a result, current theoretical analyses of LoRA fall into one of three categories. First, there are several works that examine LoRA in infinite regimes, such as its asymptotic convergence or its behavior on infinitely wide neural networks (Kim et al., 2025; Jang et al., 2024; Zeng & Lee, 2024; Yaras et al., 2024; Zhang & Pilanci, 2024; Hayou et al., 2024). However, these works do not establish a non-asymptotic convergence rate for finite models. Second, many works propose and analyze memory-efficient algorithms that resemble but do not address the original LoRA algorithm. These include GaLore (Zhao et al., 2024; He et al., 2025) and LoRA variants that only update a single adapter matrix at a time (Malinovsky et al., 2024; Sokolov et al., 2025). Third, some recent works analyze algorithmic frameworks that *do* extend to LoRA, and *do* derive a convergence rate (Jiang et al., 2024; Ghiasvand et al., 2025). However, these works require that the adapter matrices $A$ and $B$ are uniformly upper bounded by some constants which appear in the final convergence bound. These restrictive assumptions artificially enforce Lipschitz smoothness within the new parametrization, essentially reducing the problem to standard gradient descent with no substantially new proof techniques. To more align theoretical understanding with practice, we seek to answer the question:

*How fast does the original LoRA algorithm converge?*

In this work, we close the above research gap by es-

---

[1]Department of Engineering Sciences and Applied Mathematics, Northwestern University, Evanston, IL, United States [2]Department of Industrial Engineering and Management Sciences, Evanston, IL, United States. Correspondence to: Siqiao Mu <siqiaomu2026@u.northwestern.edu>.

*Proceedings of the 43rd International Conference on Machine Learning*, Seoul, South Korea. PMLR 306, 2026. Copyright 2026 by the author(s).

tablishing the first non-asymptotic convergence analysis of LoRA gradient descent, without requiring asynchronous updates, bounded adapter matrices, or Lipschitz smoothness of the reparametrized loss function. We only assume that the *original* loss function is Lipschitz smooth and lower bounded. Our novel proof approach involves stacking $B$ and $A$ into a single matrix $V$ and deriving a modified "Lipschitz-like" descent lemma for gradient descent with respect to $V$. We prove that to achieve descent at each step, the algorithm requires a learning rate that is "small enough" with respect to the norm of the current parameters and gradient. This recursive relationship between the parameters, gradient, and learning rate introduces slowdown in the convergence, resulting in an $O(\frac{1}{\log T})$ convergence rate. If we further assume that the adapter norms are bounded, we can recover the classic $O(\frac{1}{T})$ convergence rate of gradient descent on Lipschitz smooth functions.

By analyzing LoRA under minimal assumptions, we can make rigorous observations about its behavior. First, the convergence rate of LoRA does not (and should not) depend on the chosen rank, since gradient descent is a dimension-free algorithm. Second, we formalize the complex relationship between the parameter norm and the LoRA training dynamics, which has been alluded to in past works. We find that LoRA displays a "position dependency," in that the convergence is slowed if the iterates are moving away from the origin and accelerated if they are moving towards the origin. This is due to the geometric changes introduced by the LoRA reparametrization, including the creation of a stationary point at the origin *regardless* of the structure of the original loss function.

Motivated by our theoretical insights, we perform experiments to examine the impact of the learning rate under the LoRA reparametrization in practice. We train logistic regression and ResNet-18 models on the CIFAR-10 images dataset with *adaptive* and *normalized* learning rates that directly stem from theory, and we compare their performance against constant learning rates of similar values. We find that adjusting the learning rate based on the parameter norm or gradient norm can both accelerate and stabilize training by accounting for the unique structure induced by the low-rank reparametrization. Extending to the LLM setting, we observe that in high dimensions, the adaptive learning rates are beneficial for small parameter norms, but they behave close to constant learning rates for large parameter norms.

Our contributions can be summarized as follows.

- We prove for the first time that the minimum gradient norm under LoRA gradient descent converges to zero at rate $O(\frac{1}{\log T})$, only assuming that the *original loss function* is Lipschitz smooth and lower bounded.

- Based on our theory, we propose and empirically vali-

date practical methods for LoRA learning rate selection that demonstrate improved convergence.

In Section 2, we review the related work in detail and contextualize our contribution. In Section 3, we provide our convergence analysis, establishing preliminaries in Section 3.1, providing our main results in Section 3.2 and discussion in Section 3.3. Finally, in Section 4, we provide the results of our experiments. Code is open-sourced at the Github repository https://github.com/siqiaomu/lora.

## 2. Related Work

The LoRA algorithm, first proposed in (Hu et al., 2022), is motivated by the idea of a viable "low-rank representation" of deep neural networks (Oymak et al., 2019; Li et al., 2018). In particular, (Aghajanyan et al., 2021) argues that large overparametrized models reside on a lower "intrinsic dimension," and fine-tuning with a low dimension reparametrization, such as a random subset of the weight matrices, can yield strong performance while minimizing computational expenses.

LoRA is simple, popular, and empirically effective for fine-tuning large language models. Many variants have been proposed to improve the initialization, performance, memory efficiency, or privacy of the algorithm (Shen, 2025; Büyükakyüz, 2024; Li et al., 2025; Meng et al., 2024; Wang et al., 2024), including but not limited to federated LoRA (Yang et al., 2025; Sun et al., 2024; Park & Klabjan, 2025), quantized LoRA (QLoRA) (Dettmers et al., 2023), and ReLoRA (Lialin et al., 2024). While LoRA performs well in practice, its behavior in comparison to full-rank training is poorly understood, with empirical studies showing that the two approaches can produce completely different solutions (Shuttleworth et al., 2026; Biderman et al., 2024). Consequently, the theoretic properties of LoRA are highly relevant for understanding this method.

As highlighted in the introduction, the theory of LoRA convergence has been tackled from many angles. First, many works examine the behavior of LoRA in infinite regimes, whether by characterizing the kinds of solutions LoRA converges to at infinity or analyzing its convergence in the neural tangent kernel (NTK) regime, which defines the linearized training dynamics of an infinitely wide neural network (Malladi et al., 2023). For example, (Kim et al., 2025) argues that LoRA converges to a low-rank global minimum with high probability, assuming that one exists. The work (Zeng & Lee, 2024) studies the expressive power of the low-rank solutions achieved by LoRA. The work (Jang et al., 2024) shows that LoRA has no spurious local minima in the NTK regime. The LoRA+ (Hayou et al., 2024) algorithm, which updates the adapter matrices with differently scaled learning rates, is motivated by an NTK convergence

analysis. Finally, the gradient flow dynamics of LoRA have been studied specifically for matrix factorization (Xu et al., 2025). These works do not establish a convergence rate for discrete-time LoRA for general functions.

Second, there are memory-efficient LoRA-like algorithms with concrete convergence analyses that do not apply to the original LoRA formulation. These include GaLore, an algorithm which projects the *gradient* into a low-rank subspace but maintains a full-rank update of the parameters (Zhao et al., 2024; He et al., 2025), LDAdam (Robert et al., 2025), which performs adaptive optimization steps within lower dimensional subspaces, and Randomized Subspace Optimization (RSO) (Chen et al., 2025). The LoRA variant LoRA-One incorporates information from the full gradient at initialization (Zhang et al., 2025), and it is analyzed only for Gaussian data and mean-squared loss. Finally, the variants RAC-LoRA (Malinovsky et al., 2024) and Bernoulli-LoRA (Sokolov et al., 2025) only train one adapter matrix while freezing the other as a fixed projection matrix, essentially preserving Lipschitz smoothness while failing to capture the complex training dynamics of actual LoRA implementations. In practice, the LoRA matrices are updated *simultaneously*, causing nonlinearity and nonsmoothness in the optimized variables. As the original LoRA formulation is much more commonly used in practice, these analyses do not address practical implementations.

Third, several works prove the convergence of LoRA under a strong boundedness assumption that artificially enforces Lipschitz smoothness. This includes analyses of federated LoRA algorithms (Park & Klabjan, 2025; Ghiasvand et al., 2025), for which single-device LoRA is a special case. These works assume that the adapter matrix norms are uniformly bounded by constants. In addition, (Jiang et al., 2024) analyzes the convergence of LoRA assuming that the singular values of $A$ and $B$ are uniformly upper bounded, which upper bounds the matrix norms by rank-dependent constants that appear in the final convergence result.

Our work closes the gap in the existing literature by deriving the explicit convergence rate of the *original* LoRA algorithm for general models, without requiring bounded parameter norms or Lipschitz smoothness of the reparametrized loss function.

## 3. Convergence Analysis

### 3.1. Preliminaries

For real-valued matrices $M, N \in \mathbb{R}^{m \times n}$, we denote the Frobenius inner product $\langle \cdot, \cdot \rangle_F$ as

$$\langle M, N \rangle_F = \sum_{i,j} m_{ij} n_{ij} = Tr(M^T N),$$

and the Frobenius matrix norm $\|\cdot\|_F$ as

$$\|M\|_F = \sqrt{\sum_{i,j} m_{ij}^2} = \sqrt{Tr(M^T M)}.$$

The Frobenius inner product and norm for matrices are analogous to the Euclidean inner product and norm for vectors. To simplify notation, we also use $\|\cdot\|$ to denote the Frobenius matrix norm.

We can characterize fine-tuning as the model-agnostic minimization problem

$$\min_{W \in \mathbb{R}^{m \times n}} \ell(W_0 + W), \tag{1}$$

where $\ell : \mathbb{R}^{m \times n} \to \mathbb{R}$ represents the loss function, $W_0 \in \mathbb{R}^{m \times n}$ represents a frozen (pretrained) weight matrix, and $W \in \mathbb{R}^{m \times n}$ (sometimes denoted as $\Delta W$) represents the update after fine-tuning. During finetuning, only $W$ is optimized while $W_0$ remains fixed. To simplify notation, we reformulate (1) with the function $\mathcal{L}(W) = \ell(W_0 + W)$ to obtain

$$\min_{W \in \mathbb{R}^{m \times n}} \mathcal{L}(W).$$

We herein call $\mathcal{L} : \mathbb{R}^{m \times n} \to \mathbb{R}$ the *original loss function*. We note that the gradient of $\mathcal{L}$ takes matrix form as $\nabla \mathcal{L}(W) \in \mathbb{R}^{m \times n}$. In the LoRA algorithm, we parametrize $W = BA$, where $B \in \mathbb{R}^{m \times r}$, $A \in \mathbb{R}^{r \times n}$ are the low-rank *adapter* matrices with rank $r < \min\{m, n\}$. This gives the following minimization problem,

$$\min_{B \in \mathbb{R}^{m \times r}, A \in \mathbb{R}^{r \times n}} \mathcal{L}(BA). \tag{2}$$

While any kind of optimization can be applied to (2), in this work we consider the prevailing case of LoRA gradient descent, where at time step $t$ the matrices $A$, $B$ are updated simultaneously as follows,

$$
\begin{aligned}
A_{t+1} &= A_t - \eta_t \nabla_A \mathcal{L}(B_t A_t), \\
B_{t+1} &= B_t - \eta_t \nabla_B \mathcal{L}(B_t A_t),
\end{aligned}
\tag{3}
$$

where $\eta_t$ denotes the learning rate at time $t$. We require the following standard assumptions for our analysis.

**Assumption 3.1** (Lipschitz smoothness)**.** The original loss function $\mathcal{L} : \mathbb{R}^{m \times n} \to \mathbb{R}$ is differentiable, and there exists a constant $L \geq 1$ such that for all $W, W' \in \mathbb{R}^{m \times n}$,

$$\left\| \nabla \mathcal{L}(W) - \nabla \mathcal{L}(W') \right\|_F \leq L \left\| W - W' \right\|_F. \tag{4}$$

Equivalently, $\mathcal{L}$ satisfies the following *descent lemma*,

$$\mathcal{L}(W') \leq \mathcal{L}(W) + \langle \nabla \mathcal{L}(W), W' - W \rangle_F + \frac{L}{2} \|W - W'\|_F^2. \tag{5}$$

**Assumption 3.2.** The original loss function $\mathcal{L}$ is lower bounded by a constant $\mathcal{L}^*$ such that for all $W \in \mathbb{R}^{m \times n}$, $\mathcal{L}(W) \geq \mathcal{L}^*$.

Assumptions 3.1 and 3.2 are fundamental to the study of gradient descent convergence. Critically, even with these assumptions, $\mathcal{L}$ is *not* Lipschitz smooth in $B$ or $A$, preventing the analysis of (3) via classic optimization techniques.

### 3.2. Results

In the following, we detail three steps for proving convergence of LoRA. First, we reformulate the problem (2) as optimization over a single variable $V$ that contains the stacked matrices $B$ and $A^T$. We can rewrite the loss function in terms of the outer product $VV^T$. Second, we derive a modified descent lemma, analogous to the descent lemma for Lipschitz smooth functions (5), that enables descent in one step as long as the learning rate is small enough with respect to the parameter norm $\|V\|$ and gradient norm. The last step of the proof is ensuring that the learning rate does not decrease too quickly, thereby guaranteeing convergence.

**Step 1: Restructure problem into $VV^T$ form.** We stack the adapter matrices $A^T$, $B$, into a single variable $V \in \mathbb{R}^{(m+n) \times r}$. Then the outer product $VV^T$ contains $BA$ as follows,

$$V = \begin{bmatrix} B \\ A^T \end{bmatrix}, \qquad VV^T = \begin{bmatrix} BB^T & BA \\ A^TB^T & A^TA \end{bmatrix}.$$

To recover $BA$, we need to extract the top right block from $VV^T$. Let $I_n$ represent the $n \times n$ identity matrix. Let $E_1 = \begin{bmatrix} I_m & 0_{m \times n} \end{bmatrix} \in \mathbb{R}^{m \times (m+n)}$ represent the "extractor" matrix that extracts the top $m$ rows from a $(m+n) \times (m+n)$ matrix when applied from the left, and let $E_2 = \begin{bmatrix} 0_{n \times m} & I_n \end{bmatrix}^T \in \mathbb{R}^{(m+n) \times n}$ represent the matrix that extracts the right $n$ columns when applied from the right. Then we have that

$$BA = E_1 VV^T E_2.$$

We next define the function $\mathcal{J} : \mathbb{R}^{(m+n) \times r} \to \mathbb{R}$ such that

$$\mathcal{J}(V) = \mathcal{L}(E_1 VV^T E_2) = \mathcal{L}(BA). \qquad (6)$$

By construction, LoRA gradient descent (3) is equivalent to gradient descent on $\mathcal{J}(V)$,

$$V_{t+1} = V_t - \eta_t \nabla \mathcal{J}(V_t),$$

since

$$\nabla \mathcal{J}(V) = \nabla_V [\mathcal{L}(E_1 VV^T E_2)]$$
$$= \begin{bmatrix} \nabla_B \mathcal{L}(BA) \\ \nabla_{A^T} \mathcal{L}(BA) \end{bmatrix} = \begin{bmatrix} \nabla_B \mathcal{L}(BA) \\ (\nabla_A \mathcal{L}(BA))^T \end{bmatrix}.$$

**Step 2: Descent lemma.** We consider the gradient $\nabla \mathcal{J}(V) \in \mathbb{R}^{(m+n) \times r}$, which can be computed as $d\mathcal{J} = \langle \nabla \mathcal{J}(V), dV \rangle_F$. Let $g(V) = E_1 VV^T E_2$, where $g : \mathbb{R}^{(m+n) \times r} \to \mathbb{R}^{m \times n}$. Then $\mathcal{J}(V) = \mathcal{L}(g(V))$. Let $G = \nabla \mathcal{L}(X)|_{X=E_1 VV^T E_2}$ represent the gradient of $\mathcal{L}$ evaluated at $E_1 VV^T E_2$. Then we have by the chain rule,

$$\begin{aligned} d\mathcal{J} =& \langle G, dg \rangle_F \\ =& \langle G, E_1(dVV^T + VdV^T)E_2 \rangle_F \\ =& Tr(G^T E_1(dVV^T + VdV^T)E_2) \\ =& Tr(G^T E_1 dVV^T E_2) + Tr(G^T E_1 VdV^T E_2) \\ =& Tr(V^T E_2 G^T E_1 dV) + Tr(dV^T E_2 G^T E_1 V) \\ =& \langle E_1^T GE_2^T V, dV \rangle_F + \langle E_2 G^T E_1 V, dV \rangle_F \\ =& \langle 2Sym(E_1^T GE_2^T)V, dV \rangle_F, \end{aligned}$$

where $Sym(A) = \frac{A+A^T}{2}$. So we have

$$\nabla \mathcal{J}(V) = 2Sym(E_1^T \nabla \mathcal{L}(E_1 VV^T E_2)E_2^T)V. \qquad (7)$$

We note that $V = 0$ is a stationary point because it results in $\nabla \mathcal{J}(V) = 0$. The function $\mathcal{J}$ is *not* Lipschitz smooth, due to the multiplicative factor of $V$ in $\nabla \mathcal{J}(V)$. However, we can still obtain a descent lemma (Lemma 3.3) that is analogous to the classic descent lemma of Lipschitz smooth functions (5), but with more higher-order terms.

**Lemma 3.3.** *For $\mathcal{J}(V)$ defined in (6), we have for all $V_1, V_2 \in \mathbb{R}^{(m+n) \times r}$,*

$$\begin{aligned} \mathcal{J}(V_2) \leq& \mathcal{J}(V_1) + \langle \nabla \mathcal{J}(V_1), V_2 - V_1 \rangle_F \\ &+ \sqrt{2}L\|V_2 - V_1\|^2 \|V_1\|^2 + \sqrt{2}L\|V_2 - V_1\|^3 \|V_1\| \\ &+ \frac{\sqrt{2}L}{4}\|V_2 - V_1\|^4 \\ &+ \|\nabla \mathcal{L}(E_1 V_1 V_1^T E_2)\|\|V_2 - V_1\|^2. \end{aligned}$$

See Appendix A.2 for the proof.

**Step 3: Control learning rate to achieve convergence.** Based on Lemma 3.3, we can pick $\eta_t$ small enough to minimize the higher-order terms, thereby guaranteeing descent in one step as stated next.

**Lemma 3.4.** *For any $V_t \in \mathbb{R}^{(m+n) \times r}$ and $V_{t+1} = V_t - \eta_t \nabla \mathcal{J}(V_t)$, suppose that*

$$\eta_t = \min\{\frac{1}{4\sqrt{2}L(\|V_t\|^2 + \|\nabla \mathcal{L}(E_1 V_t V_t^T E_2)\|)}, 1\}. \qquad (8)$$

*Then we have*

$$\mathcal{J}(V_{t+1}) \leq \mathcal{J}(V_t) - \frac{\eta_t}{4}\|\nabla \mathcal{J}(V_t)\|^2. \qquad (9)$$

See Appendix A.3 for the proof.

Lemma 3.4 states that to update $V_{t+1}$, we require the norm of the previous iterate $\|V_t\|$ and the gradient of the original loss function $\|\nabla\mathcal{L}(E_1 V_t V_t^T E_2)\|$. By summing (9) over all $t = 0, ..., T-1$ and telescoping sums, we determine that the minimum gradient norm over $t$ is upper bounded as

$$\min_{t=0,...,T-1}\|\nabla\mathcal{J}(V_t)\|^2 \le \frac{4(\mathcal{J}(V_0) - \mathcal{L}^*)}{\sum_{t=0}^{T-1}\eta_t}. \quad (10)$$

However, (10) is *not* sufficient to conclude convergence; we also need to show that the sum $\sum_{t=0}^{T-1}\eta_t$ *diverges* as $T$ goes to infinity rather than converging to a finite value. Based on (8), this automatically holds if $\|V_t\|^2$ is uniformly upper bounded, leading to $\sum_{t=0}^{T-1}\eta_t = \Theta(T)$. However, if the parameter norm is not bounded, then $\|V_t\|^2$ may grow to infinity. In the proof of Theorem 3.5, we show that even in the worst case, we have $\|V_t\|^2 = O(t)$. The sum $\sum_{t=0}^{T-1}\eta_t$ is therefore lower bounded by a harmonic series which is $\Theta(\log(T))$, leading to Theorem 3.5.

**Theorem 3.5.** *Suppose Assumptions 3.1 and 3.2 hold. We perform $T$ steps of LoRA gradient descent (3), where at time step $t$ the learning rate $\eta_t$ is set as (8). Then after $T$ steps, we have*

$$\min_{t=0,...,T-1}\|\nabla\mathcal{J}(V_t)\|^2 = O\left(\frac{1}{\log T}\right), \quad (11)$$

*where the $O(\cdot)$ notation hides dependence on $V_0$, $\mathcal{J}(V_0)$, $\mathcal{L}^*$, and $L$. Moreover, if there exists a constant $C > 0$ such that for all $V_t$, $\|V_t\| \le C$, then*

$$\min_{t=0,...,T-1}\|\nabla\mathcal{J}(V_t)\|^2 = O\left(\frac{1}{T}\right), \quad (12)$$

*where the $O(\cdot)$ notation hides dependence on $\mathcal{J}(V_0)$, $\mathcal{L}^*$, $L$, and $C$.*

See Appendix A.4 for the proof.

Theorem 3.5 states that LoRA gradient descent converges at $O(\frac{1}{\log T})$ rate. If $\|V_t\|$ is uniformly bounded, the rate improves to $O(1/T)$. This matches the standard convergence rate of gradient descent on Lipschitz smooth functions *and* the rates derived by existing works that analyze LoRA under the bounded parameter assumption, which essentially forces $\mathcal{J}$ to be Lipschitz smooth. Our results demonstrate that the behavior of LoRA is governed by complex interactions between $\eta_t$, $\|V_t\|$, and $\|\nabla\mathcal{J}(V_t)\|$, which slows convergence. However, even if $\|V_t\|$ grows and forces $\eta_t$ to decrease at each iteration, the algorithm still converges; a simple explanation is that if $\|V_t\|$ grows, $\eta_t$ decreases, which slows the growth of $\|V_t\|$.

**Extension to multiple matrices.** The above theory assumes the loss functions is parametrized by a single weight matrix $W$. In practice, deep neural networks are parametrized by multiple weight matrices of potentially different sizes, each with its own LoRA adapter matrices. Our analysis naturally extends to this setting, which we illustrate (without loss of generality) with the following example of two matrices. Let $\mathcal{L}(W_1, W_2) \in \mathbb{R}$ denote the loss as a function of $W_1 \in \mathbb{R}^{m_1 \times n_1}$ and $W_2 \in \mathbb{R}^{m_2 \times n_2}$. For any matrix $M \in \mathbb{R}^{(m_1+m_2)\times(n_1+n_2)}$, define the linear operators $E_{11}(M) \in \mathbb{R}^{m_1 \times n_1}$ and $E_{22}(M) \in \mathbb{R}^{m_2 \times n_2}$ as follows,

$$E_{11}(M) = \begin{bmatrix} I_{m_1} & 0_{m_1 \times m_2} \end{bmatrix} M \begin{bmatrix} I_{n_1} \\ 0_{n_2 \times n_1} \end{bmatrix},$$

$$E_{22}(M) = \begin{bmatrix} 0_{m_2 \times m_1} \times I_{m_2} \end{bmatrix} M \begin{bmatrix} 0_{n_1 \times n_2} \\ I_{n_2} \end{bmatrix},$$

where $E_{11}$ extracts the top left $m_1 \times n_1$ block and $E_{22}$ extracts the bottom right $m_2 \times n_2$ block of $M$. We can define a new function $\tilde{\mathcal{L}} : \mathbb{R}^{(m_1+m_2)\times(n_1+n_2)} \to \mathbb{R}$ such that

$$\tilde{\mathcal{L}}(M) = \mathcal{L}(E_{11}(M), E_{22}(M)),$$

and the gradient $\nabla\tilde{\mathcal{L}}(M)$ is equal to the following matrix,

$$\begin{bmatrix} \nabla_{W_1}\mathcal{L}(E_{11}(M), E_{22}(M)) & 0 \\ 0 & \nabla_{W_2}\mathcal{L}(E_{11}(M), E_{22}(M)) \end{bmatrix}.$$

This follows from the fact that by construction, the terms in the top right and bottom left blocks of $M$ have no impact on the value of $\tilde{\mathcal{L}}$, and their derivatives are zero.

Overloading notation, we define the $\ell^2$ product norm on $\mathbb{R}^{m_1 \times n_1} \times \mathbb{R}^{m_2 \times n_2}$ as follows, for matrices $W_1 \in \mathbb{R}^{m_1 \times n_1}$ and $W_2 \in \mathbb{R}^{m_2 \times n_2}$,

$$\|(W_1, W_2)\|_F^2 = \|W_1\|_F^2 + \|W_2\|_F^2.$$

Then we can show that if $\mathcal{L}$ is Lipschitz smooth, $\tilde{\mathcal{L}}$ is also Lipschitz smooth.

**Lemma 3.6.** *Suppose $\mathcal{L}$ is $L$-Lipschitz smooth such that for all $W_1, W_1' \in \mathbb{R}^{m_1 \times n_1}$, and $W_2, W_2' \in \mathbb{R}^{m_2 \times n_2}$,*

$$\|\nabla\mathcal{L}(W_1, W_2) - \nabla\mathcal{L}(W_1', W_2')\| \le L\|(W_1 - W_1', W_2 - W_2')\|.$$

*Then $\tilde{\mathcal{L}}$ is $2L$-Lipschitz smooth.*

See Appendix A.5 for the proof.

Finally, we reparametrize $W_1 = B_1 A_1$ and $W_2 = B_2 A_2$, where $B_1 \in \mathbb{R}^{m_1 \times r}$, $A_1 \in \mathbb{R}^{r \times n_1}$, $B_2 \in \mathbb{R}^{m_2 \times r}$, $A_2 \in \mathbb{R}^{r \times n_1}$ represent the LoRA adapter matrices. We can construct the larger low-rank matrices $B = [B_1^T, B_2^T]^T$ and $A = [A_1, A_2]$, such that

$$M = BA = \begin{bmatrix} B_1 A_1 & B_1 A_2 \\ B_2 A_1 & B_2 A_2 \end{bmatrix}.$$

Then $B_1 A_1 = E_{11}(BA)$ and $B_2 A_2 = E_{22}(BA)$, and gradient descent on $\tilde{L}(BA)$ with respect to $B$ and $A$ is equivalent to LoRA gradient descent on $\mathcal{L}$. Since $\tilde{L}$ is Lipschitz smooth, we can directly apply the existing analysis to this setting.

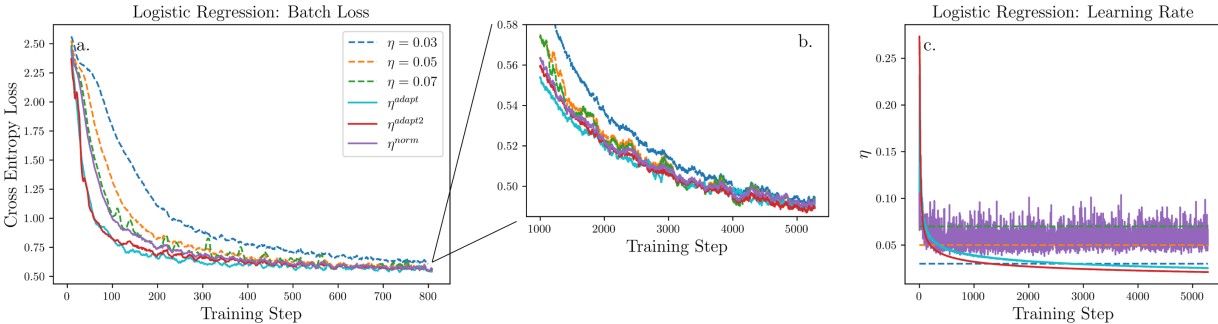

*Figure 1.* Training logistic regression model on embeddings of the CIFAR-10 dataset with constant, adaptive ($\eta^{adapt}$ and $\eta^{adapt2}$), and normalized ($\eta^{norm}$) learning rates. For clarity, the moving averages (window size 10 and 200 for a. and b. respectively) of the batch loss are plotted.

### 3.3. Discussion

Our theoretical results yield several interesting insights into the behavior of LoRA. First, the choice of extractor matrices $E_1$, $E_2$, has almost no impact on the proof, suggesting that other subsets of the $VV^T$ matrix could be extracted as alternate parameter-efficient reparametrizations to yield similar results. Moreover, the $VV^T$ form is exactly the symmetric Burer-Monteiro parametrization (Burer & Monteiro, 2003), and the results of our analysis can be trivially extended to show that gradient descent with Burer-Monteiro also converges to stationary points for general Lipschitz smooth functions. Second, the convergence rate does not explicitly depend on the chosen LoRA rank $r$, except that it is $O(\frac{1}{\|V_0\|^2})$. This is expected because $r$ just controls the dimension of the problem, but gradient descent is a *dimension-free* algorithm. Finally, due to the $\|V_t\|^2$ term in the denominator of $\eta_t$, the convergence of LoRA is slowed if the iterates are progressing away from the origin, and it is accelerated if they are moving towards the origin. This "position dependency" is not typical of gradient algorithms, and may be due to the fact that the LoRA reparametrization restructures the geometry of the loss landscape, including creating a stationary point $\nabla \mathcal{J}(V) = 0$ at $V = 0$ *regardless* of the structure of the original function (7). Consequently, it is possible (but not guaranteed) for LoRA to converge to the origin even if the original global minima is arbitrarily far away, exemplifying how LoRA and full-rank fine-tuning might lead to different outcomes.

## 4. Experiments

For additional experimental details, see Appendix B.

We conduct experiments to investigate how the learning rate and LoRA reparametrization affects training convergence in practice. We perform image classification on the CIFAR-10 dataset (Krizhevsky, 2009) with 10 classes, using the cross-entropy loss function. We first consider logistic regression, since the loss function is known to be Lipschitz smooth, and we train the model on embeddings of the CIFAR-10 images, produced by a ResNet-18 model (He et al., 2016) pretrained on ImageNet-1k (Deng et al., 2009). We then train a ResNet-18 model directly on the CIFAR-10 dataset, with LoRA implemented on the convolutional layers, and batch normalization layers disabled because they effectively change the learning rate at every step. For each experimental setting, we replace the model weight matrices with low-rank approximations ($r = 4$ and $r = 20$) and we train with mini-batch SGD with batch size $b = 512$. The model weights are first initialized randomly and frozen, and then the adapter matrices are initialized in the standard approach with $A$ as a random matrix, and $B$ as a zero matrix.

We test three types of learning rate schemes: constant, *adaptive*, and *normalized*. We first consider an adaptive learning rate $\eta^{adapt}$ defined as follows,

$$\eta_t^{adapt} = \frac{\alpha}{\|V_t\|^2 + \|\nabla\mathcal{L}(E_1 V_t V_t^T E_2)\|}.$$

This learning rate mirrors the relationship between $\eta_t$, the parameter norm $\|V_t\|$ and the intermediate gradient norm $\|\nabla\mathcal{L}(E_1 V_t V_t^T E_2)\|$ derived in (8), multiplied by a scaling factor $\alpha$. While $\eta^{adapt}$ closely reflects theory, the quantity $\|\nabla\mathcal{L}(E_1 V_t V_t^T E_2)\|$ is typically not computed in standard LoRA implementations, which save memory by computing the lower-dimensional term $x A^T B^T$ instead of $x(BA)^T$, for a row vector input $x$. We can make this approach more practical by directly using the loss value instead of the gradient, reflecting the analysis used to lower bound $\eta_t$ (See (23) in Appendix A.4). This leads to our second adaptive learning rate $\eta_t^{adapt2}$, defined as follows,

$$\eta_t^{adapt2} = \frac{\alpha}{\|V_t\|^2 + \sqrt{\mathcal{L}(E_1 V_t V_t^T E_2)}}.$$

Finally, we also consider a *normalized* learning rate $\eta_{norm}$, defined as follows,

$$\eta_t^{norm} = \frac{\alpha}{\|\nabla\mathcal{J}(V_t)\|^{1/2}}.$$

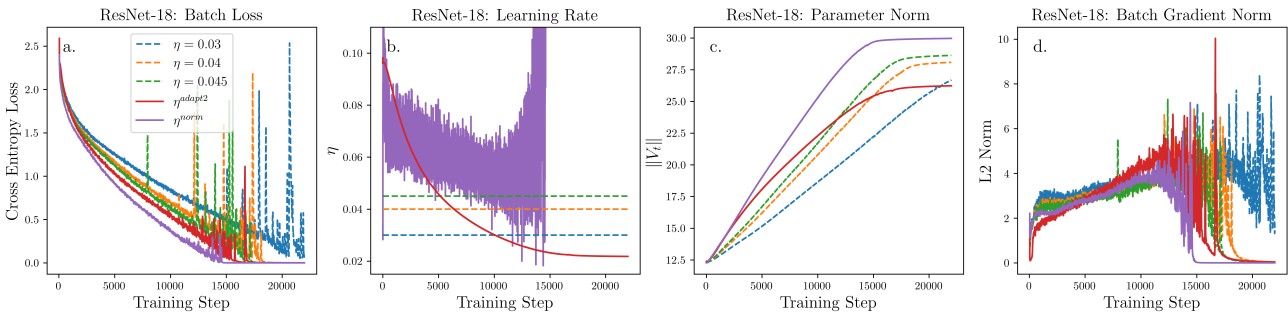

*Figure 2.* Training ResNet-18 model on the CIFAR-10 dataset with constant, adaptive ($\eta^{adapt2}$) and normalized ($\eta^{norm}$) learning rates. For clarity, the moving averages (window size 50) of the batch loss and gradient norm are plotted.

This inverse relationship between $\eta_t$ and $\|\nabla \mathcal{J}(V_t)\|^{1/2}$ is required for minimizing higher order terms in the descent lemma, as stated in (21) in Appendix A.3. In practice, we compute $\eta_t^{adapt}$, $\eta_t^{adapt2}$ and $\eta_t^{norm}$ using the *batch* gradient and loss values at training step $t$. We treat $\alpha$ as a tunable hyperparameter, and the constant learning rates are picked based on a cyclical learning rate finder (see Appendix B).

Figure 1 displays the results of the logistic regression experiments. Figures 1a and 1b demonstrate that training with the nonconstant learning rates $\eta_t^{adapt}$, $\eta_t^{adapt2}$, or $\eta_t^{norm}$ exhibits faster and more stable convergence than training with a constant learning rate in the same range of values. If the constant learning rate is too high, training becomes unstable, but if it is too low, convergence slows. Figure 1c displays the values of $\eta^{adapt}$, $\eta^{adapt2}$ and $\eta^{norm}$ over the training time, demonstrating that $\eta_t^{adapt}$ and $\eta_t^{adapt2}$ are closely correlated early on, but diverge as the model converges.

Figure 2 displays the results of the ResNet-18 experiments using $\eta_t^{adapt2}$ and $\eta_t^{norm}$. The $\eta_t^{adapt}$ learning rate is not tested due to memory constraints in this setting. Both the adaptive and normalized learning rates significantly stabilize training compared to constant learning rate while maintaining fast convergence, with $\eta^{norm}$ outperforming $\eta^{adapt2}$. Therefore, $\eta^{adapt2}$ and $\eta^{norm}$ are practical and computationally efficient methods for improving LoRA convergence on neural networks.

Figures 2c and 3a display the progress of $\|V_t\|$ over time, demonstrating how the iterates may either remain in a bounded set or grow to infinity; the latter case results in a decreasing learning rate that causes the $O(1/\log T)$ slowdown. For the ResNet-18 model, the iterates converge quickly to a global minima and $\|V_t\|$ stops growing after a finite number of iterates. However, for the logistic regression model, $\|V_t\|$ monotonically increases over 60 epochs. To investigate further, we extended the logistic regression training to 1000 epochs. Figure 4 demonstrates that although the loss appears to converge early on, $\|V_t\|$ grows unbounded for all $t$. This suggests that the iterates are converging to a stationary

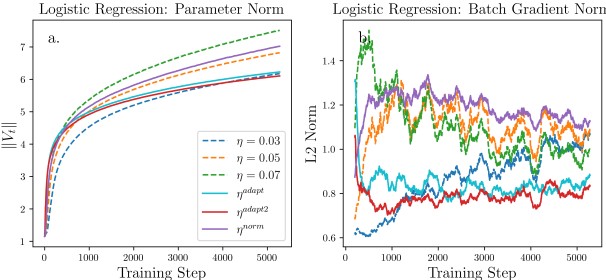

*Figure 3.* Training logistic regression model on embeddings of the CIFAR-10 dataset with constant, adaptive ($\eta^{adapt}$ and $\eta^{adapt2}$), and normalized ($\eta^{norm}$) learning rates. For clarity, the moving averages (window size 30) of the batch gradient are plotted.

point at infinity, and the convergence is indeed meaningfully slower than the standard $O(1/T)$ rate of gradient descent.

Finally, we scaled our experiments to the LLM setting. We fine-tuned TinyLlama-1.1B-Chat-v1.0 (Zhang et al., 2024) on the Alpaca dataset (Taori et al., 2023) using LoRA with a rank of 32 and batch size 16. We initialize $B$ as a zero matrix and $A$ as a Gaussian random matrix with standard deviation $\sigma$. Figure 5a and b demonstrate that if $A$ is initialized with small parameters, with $\sigma = 1e - 3$, the adaptive learning rates maintain faster and more stable convergence than constant learning rates of similar magnitudes. However, Figure 5c and d demonstrate that if the LLM is initialized with large values, with $\sigma = 1/r$, the advantage of the learning rate schedules is diminished. This is because in this high-dimensional space, $\|V_t\|$ is not just very large but also grows extremely slowly relative to its size. As a result, as shown in Figure 5d, the adaptive learning rate is close to a constant learning rate. This suggests that in this regime, the convergence may appear to the naked eye as $O(1/T)$ for some finite number of iterations, although the asymptotic behavior is slower.

Figures 3a and 2c, in combination with Figures 3b and 2d, suggest that the model is initialized in a flat region close to the origin, where the gradient norm is small. This aligns

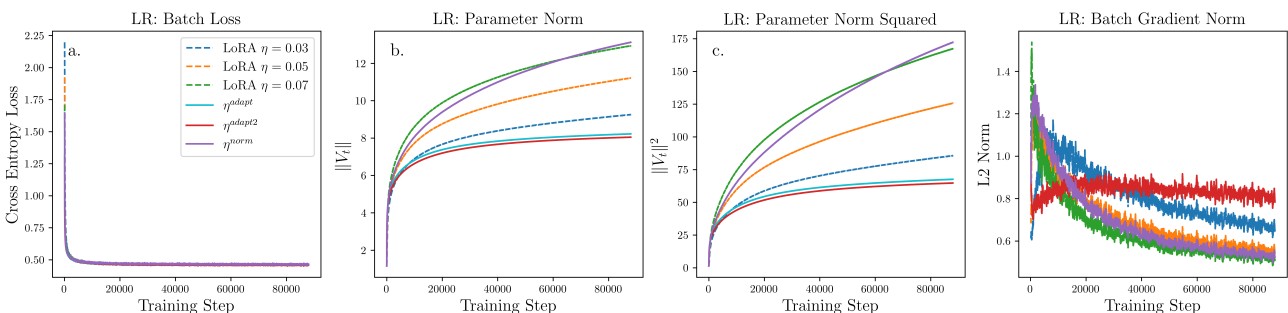

*Figure 4.* Training logistic regression model on embeddings of the CIFAR-10 dataset for 1000 epochs with constant, adaptive ($\eta^{adapt}$ and $\eta^{adapt2}$), and normalized ($\eta^{norm}$) learning rates.

with the fact that LoRA always creates a stationary point at the origin. Our results suggest that a large initial learning rate is necessary for accelerating the iterates out of this region.

Ultimately, our results demonstrate that $\eta^{adapt}$, $\eta^{adapt2}$, and $\eta^{norm}$ can improve LoRA training by taking advantage of the structural changes introduced by the low-rank reparametrization. These learning rate schemes start at higher values when the gradient norm is low and decrease over time, thereby accelerating training through the plateau around initialization and stabilizing it when the gradient norm is large. In particular, $\eta^{adapt}$ is most aligned with theory, while $\eta^{adapt2}$ and $\eta^{norm}$ are more appropriate for practical implementations.

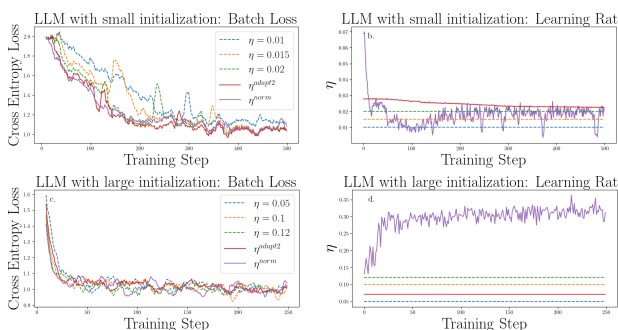

*Figure 5.* Fine-tuning LLM on Alpaca dataset with constant, adaptive ($\eta^{adapt2}$) and normalized ($\eta^{norm}$) learning rates. For clarity, the moving averages (window size 10) of the batch loss and gradient norm are plotted.

## 5. Conclusion

In this work, we show for first time that the original LoRA gradient descent algorithm achieves a convergence rate of $O(\frac{1}{\log T})$, without requiring bounded adapter matrices or Lipschitz smoothness of the reparametrized loss function.

Future research directions may include determining convergence rates on convex or strongly convex loss functions, whose geometric properties may change under the LoRA

reparametrization, as well as analyzing the *stochastic* gradient descent setting. A major challenge of this setting is deriving a fourth-moment bound on the noise.

## Impact Statement

This paper presents work whose goal is to advance the field of Machine Learning. There are many potential societal consequences of our work, none which we feel must be specifically highlighted here.

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

# A. Proofs

## A.1. Helper Lemmas

**Lemma A.1.** *(Hardy et al., 1934) (Weighted) AM-GM inequality. Given the nonnegative values $x_1, x_2, w_1, w_2$, we have*

$$\frac{w_1 x_1 + w_2 x_2}{w_1 + w_2} \geq (x_1^{w_1} x_2^{w_2})^{\frac{1}{w_1 + w_2}}. \tag{13}$$

*In particular, for $w_1 = w_2 = 1$, we have*

$$x_1 + x_2 \geq 2\sqrt{x_1 x_2}. \tag{14}$$

For the proof of Lemma A.1, see Section 2.5 of (Hardy et al., 1934).

**Lemma A.2.** *For any matrix $A \in \mathbb{R}^{(m+n) \times (m+n)}$ and the extractor matrices $E_1 = \begin{bmatrix} I_m & 0_{m \times n} \end{bmatrix}$, $E_1 \in \mathbb{R}^{m \times (m+n)}$, and $E_2 = \begin{bmatrix} 0_{n \times m} & I_n \end{bmatrix}^T$, $E_2 \in \mathbb{R}^{(m+n) \times n}$, we have*

$$\|E_1 A E_2\| \leq \|A\|.$$

*Moreover, for any matrix $B \in \mathbb{R}^{m \times n}$,*

$$\|E_1^T B E_2^T\| = \|B\|.$$

*Finally, if $A \in \mathbb{R}^{(m+n) \times (m+n)}$ is symmetric,*

$$\|E_1 A E_2\| \leq \frac{1}{\sqrt{2}} \|A\|. \tag{15}$$

*Proof.* The first statement is true because $E_1 A E_2$ contains a subset of the entries of $A$. The second statement is true because $E_1^T B E_2^T$ produces an enlarged matrix that only contains the entries of B and zeros. The last statement can be shown by observing that $E_1 A E_2$ extracts the upper right $m \times n$ block from $A$, specifically the elements at $i, j$ where $1 \leq i \leq m$ and $m + 1 \leq j \leq m + n$. So if we have

$$A = \begin{bmatrix} A_{11} & A_{12} \\ A_{12}^T & A_{22} \end{bmatrix},$$

then $\|A\|^2 = \|A_{11}\|^2 + \|A_{22}\|^2 + 2\|A_{12}\|^2$ and $\|A_{12}\| \leq \frac{1}{\sqrt{2}} \|A\|$. $\qquad \square$

**Lemma A.3.** *Let $\{a_n\}_{n \geq 0}$ be a real, nonnegative sequence, and let $c > 0$ be a fixed constant. If the series $\sum_{n=1}^{\infty} \min(a_n, c)$ converges, then the series $\sum_{n=1}^{\infty} a_n$ also converges.*

*Proof.* Let $b_n = \min(a_n, c)$. Since $\sum_{n=1}^{\infty} b_n$ converges, the terms $b_n$ must go to zero, so there exists $N > 0$ such that for $n \geq N$,

$$b_n \leq \frac{c}{2}.$$

This also implies that for $n \geq N$, $a_n \leq \frac{c}{2} < c$. So we have that $\sum_{n=N}^{\infty} a_n = \sum_{n=N}^{\infty} b_n$, and both these tail sums must converge. Then our original series can be written in terms of a finite sum and the tail sum, both of which are finite:

$$\sum_{n=1}^{\infty} a_n = \sum_{n=1}^{N-1} a_n + \sum_{n=N}^{\infty} a_n = \sum_{n=1}^{N-1} a_n + \sum_{n=N}^{\infty} b_n.$$

$\qquad \square$

## A.2. Proof of Lemma 3.3

*Proof.* Based on the expression for the gradient $\nabla\mathcal{J}(V)$ in (7), we have

$$
\begin{aligned}
\nabla\mathcal{J}(V_2) - \nabla\mathcal{J}(V_1) =& 2\big[Sym(E_1^T\nabla\mathcal{L}(E_1V_2V_2^TE_2)E_2^T)V_2 - Sym(E_1^T\nabla\mathcal{L}(E_1V_1V_1^TE_2)E_2^T)V_1\big], \\
=& 2\big[Sym(E_1^T\nabla\mathcal{L}(E_1V_2V_2^TE_2)E_2^T)V_2 - Sym(E_1^T\nabla\mathcal{L}(E_1V_1V_1^TE_2)E_2^T)V_2 \\
& + Sym(E_1^T\nabla\mathcal{L}(E_1V_1V_1^TE_2)E_2^T)V_2 - Sym(E_1^T\nabla\mathcal{L}(E_1V_1V_1^TE_2)E_2^T)V_1\big], \\
=& 2\big[(Sym(E_1^T\nabla\mathcal{L}(E_1V_2V_2^TE_2)E_2^T) - Sym(E_1^T\nabla\mathcal{L}(E_1V_1V_1^TE_2)E_2^T))V_2 \\
& + Sym(E_1^T\nabla\mathcal{L}(E_1V_1V_1^TE_2)E_2^T)(V_2 - V_1)\big], \\
=& 2\big[Sym(E_1^T\nabla\mathcal{L}(E_1V_2V_2^TE_2)E_2^T - E_1^T\nabla\mathcal{L}(E_1V_1V_1^TE_2)E_2^T)V_2 \\
& + Sym(E_1^T\nabla\mathcal{L}(E_1V_1V_1^TE_2)E_2^T)(V_2 - V_1)\big], \\
=& 2\big[Sym(E_1^T(\nabla\mathcal{L}(E_1V_2V_2^TE_2) - \nabla\mathcal{L}(E_1V_1V_1^TE_2))E_2^T)V_2 \\
& + Sym(E_1^T\nabla\mathcal{L}(E_1V_1V_1^TE_2)E_2^T)(V_2 - V_1)\big]. \tag{16}
\end{aligned}
$$

If we denote $d = V_2 - V_1$, and parametrize $\mathcal{J}$ between $V_1$ and $V_2$ such that for $t \in [0, 1]$, then

$$
\begin{aligned}
\phi(t) =& \mathcal{J}(V_1 + td), \\
\phi'(t) =& \langle\nabla\mathcal{J}(V_1 + td), d\rangle_F.
\end{aligned}
$$

By the fundamental theorem of calculus, we have

$$
\begin{aligned}
\mathcal{J}(V_2) - \mathcal{J}(V_1) = \phi(1) - \phi(0) =& \int_0^1 \phi'(t)dt, \\
=& \int_0^1 \langle\nabla\mathcal{J}(V_1), d\rangle_F\, dt + \int_0^1 \langle\nabla\mathcal{J}(V_1 + td) - \nabla\mathcal{J}(V_1), d\rangle_F\, dt, \\
=& \langle\nabla\mathcal{J}(V_1), d\rangle_F + \int_0^1 \langle\nabla\mathcal{J}(V_1 + td) - \nabla\mathcal{J}(V_1), d\rangle_F\, dt, \\
\leq& \langle\nabla\mathcal{J}(V_1), d\rangle_F + \int_0^1 \|\nabla\mathcal{J}(V_1 + td) - \nabla\mathcal{J}(V_1)\|\|d\|dt, \tag{17}
\end{aligned}
$$

where in the last step we use the Cauchy-Schwarz inequality and the fact that integrals preserve inequalities. From (16), we have

$$
\begin{aligned}
\|\nabla\mathcal{J}(V_1 + td) - \nabla\mathcal{J}(V_1)\| =& 2\|Sym(E_1^T(\nabla\mathcal{L}(E_1(V_1 + td)(V_1 + td)^TE_2) - \nabla\mathcal{L}(E_1V_1V_1^TE_2))E_2^T)(V_1 + td) \\
& + Sym(E_1^T\nabla\mathcal{L}(E_1V_1V_1^TE_2)E_2^T)td\|, \\
\overset{\text{Cauchy-Schwarz}}{\leq}& 2\|Sym(E_1^T(\nabla\mathcal{L}(E_1(V_1 + td)(V_1 + td)^TE_2) - \nabla\mathcal{L}(E_1V_1V_1^TE_2))E_2^T)\|\|V_1 + td\| \\
& + 2\|Sym(E_1^T\nabla\mathcal{L}(E_1V_1V_1^TE_2)E_2^T)\|\|td\|, \\
\overset{\|Sym(A)\|\leq\|A\|}{\leq}& 2\|E_1^T(\nabla\mathcal{L}(E_1(V_1 + td)(V_1 + td)^TE_2) - \nabla\mathcal{L}(E_1V_1V_1^TE_2))E_2^T\|\|V_1 + td\| \\
& + 2\|E_1^T\nabla\mathcal{L}(E_1V_1V_1^TE_2)E_2^T\|\|td\|, \\
\overset{\text{Lemma A.2}}{\leq}& 2\|\nabla\mathcal{L}(E_1(V_1 + td)(V_1 + td)^TE_2) - \nabla\mathcal{L}(E_1V_1V_1^TE_2)\|\|V_1 + td\| \\
& + 2t\|\nabla\mathcal{L}(E_1V_1V_1^TE_2)\|\|d\|, \\
\overset{\text{Assumption 3.1}}{\leq}& 2L\|E_1(V_1 + td)(V_1 + td)^TE_2 - E_1V_1V_1^TE_2\|\|V_1 + td\| + 2t\|\nabla\mathcal{L}(E_1V_1V_1^TE_2)\|\|d\|,
\end{aligned}
$$

$$\overset{(15)}{\leq} \sqrt{2}L\|(V_1+td)(V_1+td)^T - V_1V_1^T\|\|V_1+td\| + 2t\|\nabla\mathcal{L}(E_1V_1V_1^TE_2)\|\|d\|,$$

$$= \sqrt{2}L\|(V_1+td)(td)^T + tdV_1^T\|\|V_1+td\| + 2t\|\nabla\mathcal{L}(E_1V_1V_1^TE_2)\|\|d\|,$$

$$\leq \sqrt{2}Lt\|d\|\|V_1+td\|^2 + \sqrt{2}Lt\|d\|\|V_1\|\|V_1+td\| + 2t\|\nabla\mathcal{L}(E_1V_1V_1^TE_2)\|\|d\|,$$

$$\leq \sqrt{2}Lt\|d\|\|V_1+td\|^2 + \sqrt{2}Lt\|d\|\|V_1\|^2 + \sqrt{2}Lt^2\|d\|^2\|V_1\| + 2t\|\nabla\mathcal{L}(E_1V_1V_1^TE_2)\|\|d\|.$$

We plug this expression into the integral in (17) and pull out terms that do not depend on $t$ to obtain

$$\int_0^1 \|\nabla\mathcal{J}(V_1+td) - \nabla\mathcal{J}(V_1)\|\|d\|dt \leq \sqrt{2}L\|d\|^2\int_0^1 t\|V_1+td\|^2dt + \sqrt{2}L\|d\|^2\|V_1\|^2\int_0^1 tdt$$

$$+ \sqrt{2}L\|d\|^3\|V_1\|\int_0^1 t^2dt + 2\|\nabla\mathcal{L}(E_1V_1V_1^TE_2)\|\|d\|^2\int_0^1 tdt,$$

$$= \sqrt{2}L\|d\|^2(\frac{1}{2}\|V_1\|^2 + \frac{2}{3}\langle V_1,d\rangle_F + \frac{1}{4}\|d\|^2) + \frac{\sqrt{2}}{2}L\|d\|^2\|V_1\|^2$$

$$+ L\frac{\sqrt{2}}{3}\|d\|^3\|V_1\| + \|\nabla\mathcal{L}(E_1V_1V_1^TE_2)\|\|d\|^2,$$

$$\overset{\text{Cauchy-Schwarz}}{\leq} \frac{\sqrt{2}}{2}L\|d\|^2\|V_1\|^2 + \frac{2\sqrt{2}}{3}L\|d\|^3\|V_1\| + \frac{\sqrt{2}}{4}L\|d\|^4 + \frac{\sqrt{2}}{2}L\|d\|^2\|V_1\|^2$$

$$+ L\frac{\sqrt{2}}{3}\|d\|^3\|V_1\| + \|\nabla\mathcal{L}(E_1V_1V_1^TE_2)\|\|d\|^2,$$

$$= \sqrt{2}L\|d\|^3\|V_1\| + \frac{\sqrt{2}}{4}L\|d\|^4 + \sqrt{2}L\|d\|^2\|V_1\|^2 + \|\nabla\mathcal{L}(E_1V_1V_1^TE_2)\|\|d\|^2.$$

We therefore have for all $V_1, V_2$,

$$\mathcal{J}(V_2) \leq \mathcal{J}(V_1) + \langle\nabla\mathcal{J}(V_1), V_2-V_1\rangle_F + \sqrt{2}L\|V_2-V_1\|^3\|V_1\| + \sqrt{2}L\|V_2-V_1\|^2\|V_1\|^2$$

$$+ \frac{\sqrt{2}}{4}L\|V_2-V_1\|^4 + \|\nabla\mathcal{L}(E_1V_1V_1^TE_2)\|\|V_2-V_1\|^2.$$

$\square$

### A.3. Proof of Lemma 3.4

*Proof.* We apply the descent lemma (Lemma 3.3) with $V_1 = V_t$, $V_2 = V_{t+1} = V_t - \eta_t\nabla\mathcal{J}(V_t)$, which yields

$$\mathcal{J}(V_{t+1}) \leq \mathcal{J}(V_t) - \eta_t\langle\nabla\mathcal{J}(V_t), \nabla\mathcal{J}(V_t)\rangle_F + \sqrt{2}L\eta_t^3\|\nabla\mathcal{J}(V_t)\|^3\|V_t\| + \sqrt{2}L\eta_t^2\|\nabla\mathcal{J}(V_t)\|^2\|V_t\|^2$$

$$+ \frac{\sqrt{2}}{4}\eta_t^4L\|\nabla\mathcal{J}(V_t)\|^4 + \eta_t^2\|\nabla\mathcal{L}(E_1V_tV_t^TE_2)\|\|\nabla\mathcal{J}(V_t)\|^2,$$

$$= \mathcal{J}(V_t) - \eta_t\|\nabla\mathcal{J}(V_t)\|^2 + \sqrt{2}L\eta_t^3\|\nabla\mathcal{J}(V_t)\|^3\|V_t\| + \eta_t^2\|\nabla\mathcal{J}(V_t)\|^2(\sqrt{2}L\|V_t\|^2 + \|\nabla\mathcal{L}(E_1V_tV_t^TE_2)\|)$$

$$+ \frac{\sqrt{2}L}{4}\eta_t^4\|\nabla\mathcal{J}(V_t)\|^4.$$

We want to select $\eta_t$ to minimize the last three terms, such that they sum to a value smaller than $\eta_t\|\nabla\mathcal{J}(V_t)\|^2$. This will guarantee descent in function value $\mathcal{J}$. Let

$$\eta_t = \min\{\frac{1}{4\sqrt{2}L(\|V_t\|^2 + \|\nabla\mathcal{L}(E_1V_tV_t^TE_2)\|)}, 1\}.$$

In the following, we prove that $\eta_t$ is smaller than various upper bounds that will allow us to achieve a clean descent result. First, since $L \geq 1$, we have the following bound,

$$\eta_t \leq \frac{1}{4(\sqrt{2}L\|V_t\|^2 + \sqrt{2}L\|\nabla\mathcal{L}(E_1V_tV_t^TE_2)\|)} \leq \frac{1}{4(\sqrt{2}L\|V_t\|^2 + \|\nabla\mathcal{L}(E_1V_tV_t^TE_2)\|)}. \tag{18}$$

For the next bound, we have from Cauchy-Schwarz and (7) that

$$\|\nabla\mathcal{J}(V)\| \leq 2\|\nabla\mathcal{L}(E_1 VV^T E_2)\|\|V\|. \tag{19}$$

Therefore, from (14) of Lemma A.1, we have $\frac{1}{a+b} \leq \frac{1}{2\sqrt{ab}}$ for $a, b > 0$, such that $\eta_t$ satisfies the following bound,

$$\eta_t \leq \frac{1}{4\sqrt{2}L\|V_t\|^2 + 4\|\nabla\mathcal{L}(E_1 V_t V_t^T E_2)\|} \leq \left(\frac{1}{64\sqrt{2}L\|V_t\|^2\|\nabla\mathcal{L}(E_1 V_t V_t^T E_2)\|}\right)^{1/2},$$

$$\overset{(19)}{\leq} \left(\frac{1}{32\sqrt{2}L\|V_t\|\|\nabla\mathcal{J}(V_t)\|}\right)^{1/2}, \tag{20}$$

$$\leq \left(\frac{1}{4\sqrt{2}L\|\nabla\mathcal{J}(V_t)\|\|V_t\|}\right)^{1/2}. \tag{21}$$

For the next bound, by (13) of Lemma A.1 with $w_1 = 1$, $w_2 = 2$, $x_1 = 4\sqrt{2}\|V_t\|^2$, $x_2 = \|\nabla\mathcal{L}(E_1 V_t V_t^T E_2)\|$, we have

$$4\sqrt{2}L\|V_t\|^2 + 2\|\nabla\mathcal{L}(E_1 V_t V_t^T E_2)\| \geq 3\left(4\sqrt{2}L\|V_t\|^2\|\nabla\mathcal{L}(E_1 V_t V_t^T E_2)\|^2\right)^{1/3},$$

$$\overset{(19)}{\geq} 3\left(\frac{4\sqrt{2}L\|\nabla\mathcal{J}(V_t)\|^2}{4}\right)^{1/3},$$

$$= 3(\sqrt{2}L\|\nabla\mathcal{J}(V_t)\|^2)^{1/3}.$$

So $\eta_t$ also satisfies the bound

$$\eta_t \leq \frac{1}{4\sqrt{2}L\|V_t\|^2 + 4\sqrt{2}L\|\nabla\mathcal{L}(E_1 V_t V_t^T E_2)\|},$$

$$\leq \frac{1}{4\sqrt{2}L\|V_t\|^2 + 2\|\nabla\mathcal{L}(E_1 V_t V_t^T E_2)\|},$$

$$\leq \left(\frac{1}{27\sqrt{2}L\|\nabla\mathcal{J}(V_t)\|^2}\right)^{1/3},$$

$$\leq \left(\frac{1}{\sqrt{2}L\|\nabla\mathcal{J}(V_t)\|^2}\right)^{1/3}. \tag{22}$$

In conclusion, with the choice of $\eta_t$ in (8), we can substitute the derived bounds on $\eta_t$ into Lemma 3.3, achieving descent in one step.

$$\mathcal{J}(V_{t+1}) \leq \mathcal{J}(V_t) - \eta_t\|\nabla\mathcal{J}(V_t)\|^2 + \overbrace{\sqrt{2}L\eta_t^3\|\nabla\mathcal{J}(V_t)\|^3\|V_t\|}^{(21)} + \overbrace{\eta_t^2\|\nabla\mathcal{J}(V_t)\|^2(\sqrt{2}L\|V_t\|^2 + \|\nabla\mathcal{L}(E_1 V_t V_t^T E_2)\|)}^{(18)}$$

$$+ \underbrace{\frac{\sqrt{2}L}{4}\eta_t^4\|\nabla\mathcal{J}(V_t)\|^4}_{(22)}$$

$$\mathcal{J}(V_{t+1}) \leq \mathcal{J}(V_t) - \eta_t\|\nabla\mathcal{J}(V_t)\|^2 + \frac{\eta_t}{4}\|\nabla\mathcal{J}(V_t)\|^2 + \frac{\eta_t}{4}\|\nabla\mathcal{J}(V_t)\|^2 + \frac{\eta_t}{4}\|\nabla\mathcal{J}(V_t)\|^2$$

$$\leq \mathcal{J}(V_t) - \frac{\eta_t}{4}\|\nabla\mathcal{J}(V_t)\|^2.$$

$\square$

## A.4. Proof of Theorem 3.5

*Proof.* From Lemma 3.4, we know that if

$$\eta_t = \min\{\frac{1}{4\sqrt{2}L(\|V_t\|^2 + \|\nabla\mathcal{L}(E_1 V_t V_t^T E_2)\|)}, 1\},$$

then we have descent in one step

$$\mathcal{J}(V_{t+1}) - \mathcal{J}(V_t) \leq -\frac{\eta_t}{4}\|\nabla\mathcal{J}(V_t)\|^2.$$

Rearranging this equation and summing on both sides yields

$$\frac{\eta_t}{4}\|\nabla\mathcal{J}(V_t)\|^2 \leq \mathcal{J}(V_t) - \mathcal{J}(V_{t+1}),$$

$$\sum_{t=0}^{T-1}\frac{\eta_t}{4}\|\nabla\mathcal{J}(V_t)\|^2 \leq \sum_{t=0}^{T-1}\mathcal{J}(V_t) - \mathcal{J}(V_{t+1}),$$

$$= \mathcal{J}(V_0) - \mathcal{J}(V_T),$$

$$\overset{\text{Assumption 3.2}}{\leq} \mathcal{J}(V_0) - \mathcal{L}^*,$$

$$\min_{t=0,\dots,T-1}\|\nabla\mathcal{J}(V_t)\|^2 \sum_{t=0}^{T-1}\eta_t \leq 4(\mathcal{J}(V_0) - \mathcal{L}^*),$$

$$\min_{t=0,\dots,T-1}\|\nabla\mathcal{J}(V_t)\|^2 \leq \frac{4(\mathcal{J}(V_0) - \mathcal{L}^*)}{\sum_{t=0}^{T-1}\eta_t}.$$

So to show convergence to a stationary point, we need to show that the sum of learning rates $\sum_{t=0}^{\infty}\eta_t$ diverges. From Lipschitz smoothness of $\mathcal{L}$, we have for all $W \in \mathbb{R}^{m\times n}$ (See Lemma 2.28 in (Garrigos & Gower, 2024)),

$$\|\nabla\mathcal{L}(W)\|^2 \leq 2L(\mathcal{L}(W) - \mathcal{L}^*), \tag{23}$$

which allows us to bound the gradient as follows,

$$\|\nabla\mathcal{L}(W)\| \leq (2L(\mathcal{L}(W) - \mathcal{L}^*))^{1/2},$$

$$\|\nabla\mathcal{L}(E_1 VV^T E_2)\| \leq (2L(\mathcal{L}(E_1 VV^T E_2) - \mathcal{L}^*))^{1/2} = (2L(\mathcal{J}(V) - \mathcal{L}^*))^{1/2}.$$

We can obtain the following lower bound, using the fact that $\mathcal{J}(V_t)$ is decreasing with $t$,

$$\frac{1}{4\sqrt{2}L(\|V_t\|^2 + \|\nabla\mathcal{L}(E_1 V_t V_t^T E_2)\|)} \geq \frac{1}{4\sqrt{2}L(\|V_t\|^2 + (2L(\mathcal{J}(V_t) - \mathcal{L}^*))^{1/2})},$$

$$\geq \frac{1}{4\sqrt{2}L(\|V_t\|^2 + (2L(\mathcal{J}(V_0) - \mathcal{L}^*))^{1/2})}.$$

Let $D_t = 4\sqrt{2}L(\|V_t\|^2 + (2L(\mathcal{J}(V_0) - \mathcal{L}^*))^{1/2})$, such that we can lower bound $\eta_t$ as follows,

$$\eta_t = \min\{\frac{1}{4\sqrt{2}L(\|V_t\|^2 + \|\nabla\mathcal{L}(E_1 V_t V_t^T E_2)\|)}, 1\} \geq \min\{\frac{1}{D_t}, 1\}.$$

To show the divergence of $\sum_{t=0}^{\infty}\eta_t$, it is sufficient to show the divergence of the series $\sum_{t=0}^{\infty}\min\{\frac{1}{D_t}, 1\}$. Moreover, by the contrapositive of Lemma A.3, we only need to show divergence of $\sum_{t=0}^{\infty}\frac{1}{D_t}$. We first show that we achieve divergence if the adapter norms are bounded. If $\|V_t\| \leq C$ for all $t$, then $\eta_t$ is lower bounded by a nonzero constant $\eta > 0$

$$\eta := \min\{\frac{1}{4\sqrt{2}L(C^2 + (2L(\mathcal{J}(V_0) - \mathcal{L}^*))^{1/2})}, 1\} \leq \eta_t,$$

and we prove (12) of Theorem 3.5 as follows

$$\min_{t=0,\dots,T-1} \|\nabla \mathcal{J}(V_t)\|^2 \leq \frac{4(\mathcal{J}(V_0) - \mathcal{L}^*)}{\eta T} = O(1/T).$$

Now we consider the general case where the adapter norms are not necessarily bounded. In the following, we show that $D_t = O(t)$. We first control the growth of $\|V_t\|^2$, showing that it increases at most linearly with $t$. We have

$$\begin{aligned}
\|V_{t+1}\|^2 =& \|V_t - \eta_t \nabla \mathcal{J}(V_t)\|^2, \\
\leq& \|V_t\|^2 + 2\eta_t |\langle V_t, \nabla \mathcal{J}(V_t)\rangle_F| + \eta_t^2 \|\nabla \mathcal{J}(V_t)\|^2, \\
\overset{\eta_t \leq 1}{\leq}& \|V_t\|^2 + 2\eta_t |\langle V_t, \nabla \mathcal{J}(V_t)\rangle_F| + \eta_t \|\nabla \mathcal{J}(V_t)\|^2, \\
\leq& \|V_t\|^2 + \eta_t(\|V_t\|^2 + \|\nabla \mathcal{J}(V_t)\|^2) + \eta_t \|\nabla \mathcal{J}(V_t)\|^2, \\
\leq& \|V_t\|^2 + \frac{\|V_t\|^2}{4\sqrt{2}L(\|V_t\|^2 + \|\nabla \mathcal{L}(E_1 V_t V_t^T E_2)\|)} + 2\eta_t \|\nabla \mathcal{J}(V_t)\|^2, \\
\leq& \|V_t\|^2 + \frac{1}{4\sqrt{2}L} + 2\eta_t \|\nabla \mathcal{J}(V_t)\|^2,
\end{aligned}$$

$$\|V_{t+1}\|^2 - \|V_t\|^2 \leq \frac{1}{4\sqrt{2}L} + 2\eta_t \|\nabla \mathcal{J}(V_t)\|^2,$$

$$\sum_{t=0}^{T-1} \|V_{t+1}\|^2 - \|V_t\|^2 \leq \frac{T}{4\sqrt{2}L} + 2\sum_{t=0}^{T-1} \eta_t \|\nabla \mathcal{J}(V_t)\|^2,$$

$$\leq \frac{T}{4\sqrt{2}L} + 8(\mathcal{J}(V_0) - \mathcal{L}^*),$$

$$\|V_T\|^2 \leq \|V_0\|^2 + \frac{T}{4\sqrt{2}L} + 8(\mathcal{J}(V_0) - \mathcal{L}^*).$$

We have that $D_t$ is upper bounded by a linear term $t$,

$$D_t = 4\sqrt{2}L(\|V_t\|^2 + (2L(\mathcal{J}(V_0) - \mathcal{L}^*))^{1/2}) \leq t + 4\sqrt{2}L(\|V_0\|^2 + 8(\mathcal{J}(V_0) - \mathcal{L}^*) + (2L(\mathcal{J}(V_0) - \mathcal{L}^*))^{1/2}).$$

We can therefore lower bound the series with a harmonic series as follows,

$$\begin{aligned}
\sum_{t=0}^{T-1} \frac{1}{D_t} \geq& \sum_{t=0}^{T-1} \frac{1}{t + 4\sqrt{2}L(\|V_0\|^2 + 8(\mathcal{J}(V_0) - \mathcal{L}^*) + (2L(\mathcal{J}(V_0) - \mathcal{L}^*))^{1/2})}, \\
=& \Theta(\log(T)).
\end{aligned}$$

Finally,

$$\min_{t=0,\dots,T-1} \|\nabla \mathcal{J}(V_t)\|^2 \leq \frac{4(\mathcal{J}(V_0) - \mathcal{L}^*)}{\sum_{t=0}^{T-1} \eta_t} = O\left(\frac{1}{\log T}\right).$$

$\square$

## A.5. Proof of Lemma 3.6

Suppose $M, M' \in \mathbb{R}^{(m_1+m_2)\times(n_1+n_2)}$. Then we have

$$
\begin{aligned}
\|\nabla \tilde{L}(M) - \nabla \tilde{L}(M')\|^2 =& \|\nabla_{W_1}\mathcal{L}(E_{11}(M), E_{22}(M)) - \nabla_{W_1}\mathcal{L}(E_{11}(M'), E_{22}(M'))\|^2, \\
& + \|\nabla_{W_2}\mathcal{L}(E_{11}(M), E_{22}(M)) - \nabla_{W_2}\mathcal{L}(E_{11}(M'), E_{22}(M'))\|^2, \\
=& \|\nabla\mathcal{L}(E_{11}(M), E_{22}(M)) - \nabla\mathcal{L}(E_{11}(M'), E_{22}(M'))\|^2, \\
\|\nabla \tilde{L}(M) - \nabla \tilde{L}(M')\| =& \|\nabla\mathcal{L}(E_{11}(M), E_{22}(M)) - \nabla\mathcal{L}(E_{11}(M'), E_{22}(M'))\|, \\
\leq& L\|(E_{11}(M), E_{22}(M)) - (E_{11}(M'), E_{22}(M'))\|, \\
=& L\|(E_{11}(M - M'), E_{22}(M - M'))\|, \\
\leq& L\|E_{11}(M - M')\| + L\|E_{22}(M - M')\| \leq 2L\|M - M'\|,
\end{aligned}
$$

where the last step follows from similar logic as Lemma A.2.

# B. Experimental Details

In the following, we provide additional details about our experimental setup. As mentioned in the paper, for the ResNet-18 experiments we turn off BatchNorm layers because they effectively change the learning rate. This is achieved by setting the model to evaluation mode with `model.eval()` before training. Table 1 displays the hyperparameters used in our experimental results. Code is open-sourced at the Github repository https://github.com/siqiaomu/lora.

*Table 1.* Experiment hyperparameters.

| Hyperparameter | Logistic regression with ResNet-18 embeddings | ResNet-18 |
|---|---|---|
| Batch size | 512 | 512 |
| LoRA rank | 4 | 20 |
| Data input dimension | 512 | $32 \times 32 \times 3$ |
| Epochs | 60 | 250 |
| $\alpha^{adapt}$ | 1 | |
| $\alpha^{adapt2}$ | 0.8 | 15 |
| $\alpha^{norm}$ | 0.06 | 0.1 |
| Constant learning rate range | [0.03, 0.05, 0.07] | [0.03, 0.04, 0.045] |

**Learning rate.** While standard LoRA implementations tend to scale by both a scaling factor and $\frac{1}{r}$, for simplicity we do not scale by the rank and just set the learning rate as a whole. To determine the range of learning rates to test in our experiments, we use a cyclical learning rate finder as introduced in (Smith, 2017) and further described in (Ahmed, 2025). We start with a small learning rate and train the model for a single epoch, repeating this with exponentially increasing learning rates. We plot the loss over the learning rates and pick learning rates in the range where the loss decreases the fastest. The resulting plots are available in Figure 6.

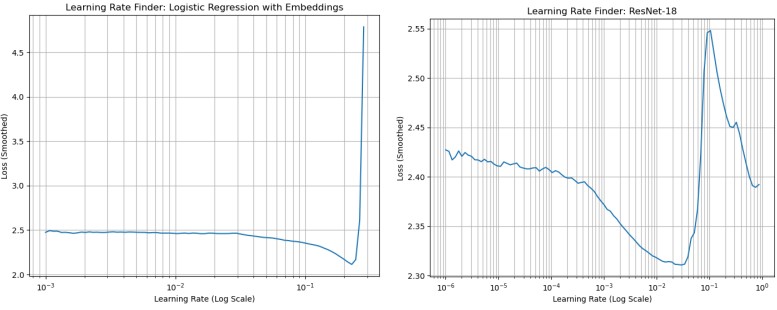

*Figure 6.* Learning rate selection using a cyclical finder for both experimental settings.

**Hardware and software.** All experiments were run using PyTorch 2.5.0 and CUDA 12.1, on an Intel(R) Xeon(R) Silver 4208 CPU (2.10GHz) with an NVIDIA RTX A6000 GPU.

**Comparison with full-rank training.** We conducted additional experiments comparing the convergence of (mini-batch) gradient descent with and without LoRA. The results in Figures 7 and 8 on logistic regression and ResNet-18 respectively support that the convergence of LoRA gradient descent is slower.

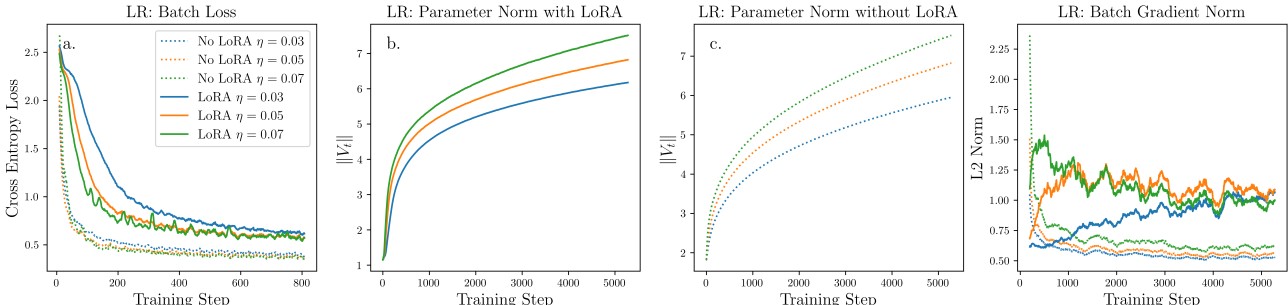

*Figure 7.* Training logistic regression model on embeddings of the CIFAR-10 dataset with and without LoRA with constant learning rates. For clarity, the moving averages (window size 10) of the batch loss and gradient norm are plotted.

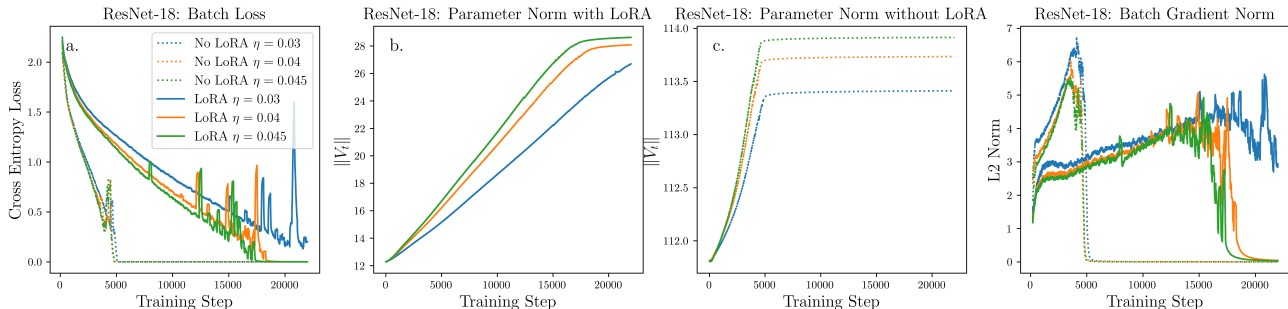

*Figure 8.* Training ResNet-18 model on the CIFAR-10 dataset with and without LoRA with constant learning rates. For clarity, the moving averages (window size 200) of the batch loss and gradient norm are plotted.

