# OpenReview forum: "On the Convergence Rate of LoRA Gradient Descent"
_ICML.cc/2026/Conference — ICML 2026 spotlight_

### Official Review · Reviewer_ZmMJ · 2026-02-25

**Soundness:** 3
**Presentation:** 3
**Significance:** 2
**Originality:** 2
**Overall Recommendation:** 5
**Confidence:** 3

**Summary:**

The paper aims to provide convergence results for LoRA, a widely used technique for paremeter efficient fine-tuning. The paper claims that they establish minimal assumptions on the original loss (i.e., Lipschitz smoothness and a uniform lower bound) compared to other papers, and still be able to obtain guarantees. The core observation of the paper is to do some lifting or recasting of the two matrices implied in LoRA into a single matrix and analyze the dynamics of gradient descent with respect to this new variable. Thus, the paper is able to obtain that the minimal gradient of the loss across $T$ iterations vanishes according to $O(1/\log(T))$ and, under stronger assumptions, $O(1/T)$. Of course, it all depends on judiciously choosing the learning rate.

**Compliance With Llm Reviewing Policy:**

Affirmed.

**Final Justification:**

I decided to increase my score based on the rebuttal that has assessed my main concerns.

**Key Questions For Authors:**

Please, see above.

**Limitations:**

yes

**Strengths And Weaknesses:**

The problem studied by the paper is highly relevant to the ML community and it is well-motivated. Another strength of the paper is that its general approach to prove its main results is easy to follow --- in fact, I believe that most people with some mathematical background will be able to grasp the paper, even if optimization is not their main expertise. The paper also provides appropriate remarks and explanations after the theoretical results.

Now, I have some concerns I want to point out to the reviewers. I will almost exclusively focus on the theoretical aspect of the paper, since their main contributions lay therein.

**Important concerns:**

- I am concerned about the claim that the paper is proving *convergence* because I do not think the paper is quite doing that. If we look at the main theorem, which is Theorem 3.5, the idea is that the quantity $\min_{t=0,\dots,T-1}||\nabla\mathcal{J}(V_t)||^2$ goes to zero as $T$ increases. However, this does not imply *convergence* in itself. This only implies that at some time $\tilde{t}$ the gradient is close to zero for large $T$ --- it does not mean that there is convergence to a stationary point as $T$ grows. For example, for a very large $T$, the time $\tilde{t}\ll T-1$ could be such that $\tilde{t}\in\arg\min_{t=0,\dots,T-1}||\nabla\mathcal{J}(V_t)||^2$ and $||\nabla\mathcal{J}(V_{\tilde{t}})||^2\approx 0$; but, at the same time, it is quite possible that $||\nabla\mathcal{J}(V_{T-1})||^2\gg 0$ --- i.e., there is no convergence as $T$ grows. The authors must show how their results truly imply convergence.

- Considering my previous point --- or in addition to it, if the authors are able to show that they are truly proving convergence --- the paper will greatly benefit from showing either of the following two *standard* convergence types: *parameter convergence* and *loss convergence*.
   - *Parameter convergence*: In the context of the paper, it would be to show that, as $T$ grows, the quantity $||V_{T-1}-V*||$ decreases at some rate, where $V*$ is a global minimum of $\mathcal{L}$, i.e., the one that produces the value $\mathcal{L}^*$ according to Assumption 3.2. The analysis may not be as "global" and may require starting close enough to a global minimum (and perhaps assume that all global minima are isolated, etc.)
   - *Loss convergence*: It is to show that $\mathcal{L}(V_{T-1}) - \mathcal{L}^*$ decreases at some rate as $T$ grows. Notice that in this case we only care about how the loss decreases --- it doesn't matter if we have global minima living in the most arbitrary manifolds.

    These types of convergence are more common in the literature of optimization/training convergence and a reason why other theoretical works (perhaps not necessarily about LoRA) need to make extra assumptions when defining convergence. The authors still has space to add the analysis of any of these types of convergence, and show how it relates to their current "minimum gradient vanishing" result (Theorem 3.5).
- I want to remark that proving parameter and loss convergence may be the reason why perhaps other works have made more assumptions. Do the authors know of any other LoRA related literature working with such convergence types? It would be *unfair* for the authors to criticize other works about having more assumptions when their convergence results are actually *stronger* --- and arguably more appropriate --- than the one by the authors.

- Lines 294-295 state that one could choose any of the submatrices from $VV^T$ to obtain other parameter-efficient reparameterizations. This is seemingly a pragmatic side of the paper's theoretical results. The problem with that statement is that such other alternatives may not be useful. For example, is it useful to simply use $BB^T$ as a parameterization?
- Lines 311-312 (as also seemingly implied in 082-086) state that the zero matrix ($V=0$), also call the *origin*, is a stationary point. I have two problems with this:
   1. This is stated without a proof, this needs to be shown even if it is seemingly simple.
   2. Once the origin is proved to be a stationary point, one needs to prove that it is attractive, i.e., that whenever the initial condition $V_0$ is sufficiently close to the origin, there will be convergence toward it.

- Derivations on the top of second column of page 4: I think there needs to be better indication of the differentiation notation. When using $d$, with respect to what is it differentiating?
- There is sloppiness and confusion regarding the use of the norm noration $||\cdot||$. Up to page 3, the only norm being defined is the Frobenius norm $||\cdot||_F$ and it is understood the paper is using such norm to define the metric space where the analysis lives (e.g., the space where differentiation operations are done in the top of second column of page 4). However, in the rest of theoretical results presented in the paper, all norms are expressed without the subscript "$F$", i.e., there is no indication that the norms are Frobenius. Are the authors using another type of norm? What norm is it used in the proofs?

(Note: I have used $||\cdot||$ instead of $||\cdot||_F$ in this review in order to express the norm used in the authors' results).


**Other concerns:**

- The paper clams that "the recursive relationship between the parameters, gradient, and learning rate introduces instability in the convergence, resulting in an $O(\frac{1}{\log T})$". However, I don't see any theoretical proof that demonstrates the presence of "instability in the convergence". Where is it? Now, it seems this is only shown numerically, in which case, this must be explicitly stated in the first column of page 2, where the quote was extracted in lines 065-068.
- There seems to be a contradiction on first column of page 2: first it says in lines 065-068 that there is instability in the convergence by the way one chooses the learning rate, but then in lines 093-097, it says that choosing the learning rate stabilizes the training. Can the authors clarify?
- I understand that the paper is dealing with matrices and working on that space instead of, say, "vectors". Thus, I know in Section 3 that the gradients have the shape as the matrix they are being taken with respect of. However, it is possible that some readers may not be familiar about thinking of gradients as matrices: I recommend a remark explaining the size of the gradients and how they are defined/computed when we are dealing with a vector space of matrices.


----

**Conclusion:** I truly hope this lengthy review helps the authors' paper. I am looking forward to the authors' response. I am giving a "weak reject" for now, but my score might change more positively or negatively depending on the authors' response.

---

**Note:** If the authors cannot fully respond to my review in a single message due to space constraints on their rebuttal, I suggest breaking their rebuttal in multiple messages/responses. If the system allows only a single response as a rebuttal and the authors still want to say more things to me, I suggest that the authors let me know about this at the end of their response --- then I can just send a placeholder message that the authors can use to continue their rebuttal by responding to it (assuming this will work).

---

> ### Author Rebuttal · Authors · 2026-03-27
>
> We thank you for your detailed review and constructive remarks.
>
> **Regarding convergence.**
>
> We believe that in the context of the existing literature regarding LoRA and nonconvex optimization in machine learning, our contribution is appropriate and our comparisons are fair. It is true that bounding $ \min_{t = 0,...,T-1} \lVert \nabla \mathcal{J}(V_t) \rVert^2$ does not imply that the *last iterate* $\lVert  \nabla \mathcal{J}(V_T) \rVert$ will converge to zero, and we will amend our wording to make that point extremely clear. It is also true that parameter or loss convergence would indeed be stronger guarantees. However, our bound, which follows from upper bounding $\sum_{t = 0}^{T-1} \eta_t \lVert \nabla J(V_t) \rVert^2$ by a constant, is a standard criteria for establishing *convergence rate* for nonconvex optimization. Moreover, because LoRA is designed for neural networks, which are highly nonconvex, stronger guarantees such as parameter or loss convergence are typically not achievable without unrealistic assumptions like convexity, strong convexity, or gradient dominance. These kinds of results are theoretically interesting but less relevant to the main use case of LoRA.
>
> Consequently, our problem setting and results match closely to that of existing literature regarding LoRA and LoRA-like algorithms. Existing works show analogous bounds in one of the following ways:
>
> * Bounding $ \min_{t = 0,...,T-1} \lVert \nabla \mathcal{J}(V_t) \rVert^2$ (Theorem 2 of Jiang et al., 2024), (Theorem 4.2 of Park & Klabjan, 2025)
>
> * Bounding the running average $\frac{1}{T} \sum_{t = 0}^{T-1}  \lVert \nabla \mathcal{J}(V_t) \rVert^2$ (Theorem 4.8 of Ghiasvand et al. 2025)
>
> * Bounding $\mathbb{E}[ \lVert \nabla \mathcal{J}(V) \rVert^2]$, where $V$ is chosen uniformly at random from all $T$ iterates and the expectation is taken over this random selection. (Theorem 5.3 Malinkovski et al., 2024), (Theorem 5.5 of Chen et al., 2025).
>
> These kinds of bounds generally do not imply last-iterate convergence, but the focus of these works is instead deriving an explicit convergence rate of the bounded metric. Moreover, none of the above works consider parameter or loss convergence, because they are far less relevant to the problem setting. Our result is therefore an appropriate point of comparison with existing work, and our main contribution is that we relax their assumptions (smoothness, bounded parameter norm) to achieve a convergence rate for the original LoRA algorithm.
>
> **Lines 294-295 and using $BB^T$ as a parameterization.**
>
> We find this feature of the analysis interesting because the original choice of BA for LoRA was somewhat arbitrary, and as such is also not consequential in the final result. It's possible that $BB^T$ (with adjustments for nonsquare matrices) could be easier to implement and use fewer parameters. We will likely pursue this research direction further.
>
> **Lines 311-312 and the origin as a stationary point.**
>
> The origin $V = 0$ is a stationary point because by the formula for $\nabla \mathcal{J}(V)$ in (7), if $V = 0$ we have $\nabla \mathcal{J}(V) = 0$. We will state this explicitly in the paper. However, it is not necessarily attractive nor are the iterates guaranteed to converge to it. We highlight this stationary point because it demonstrates how the LoRA reparametrization restructures the loss function, leading to a dependence on the distance from the origin in the convergence rate.
>
> **Derivations on the top of second column of page 4.**
>
>  $dJ$ represents the differential of $J$, not the derivative. To derive the gradient, we utilize the fact that $dJ = \langle \nabla J(V), dV \rangle_F$. As mentioned below, we will add a remark to clarify this.
>
> **"There is sloppiness and confusion regarding the use of the norm notation."**
>
> We respectfully point out that in line 117, we state that we use both $\lVert \cdot \rVert$ and $\lVert \cdot \rVert_F$ to denote the Frobenius matrix norm. However, to improve clarity, we will add the subscript $F$ to all norm notation.
>
> **Other concerns.**
>
> **Bullet points 1 and 2.** In this line 065-068, we are referring to the fact that the convergence requires a carefully controlled learning rate, which depends on the size of the parameter or gradient norm, which in turn also depend on the learning rate. If the learning rate exits this "stable" delicate regime, function descent/convergence is not guaranteed and therefore "unstable". We see that our nontechnical use of "instability" is perhaps confusing, and we will replace it with "slowdown."
>
> **Bullet point 3.** Thank you for this suggestion. We will include a remark in Section 3.1 clarifying that the gradients are indeed matrices, and can be computed via $dJ = \langle \nabla J(V), dV \rangle_F$, which is analogous to the vector definition with the Euclidean dot product $\langle \cdot, \cdot \rangle_2$.

---

> > ### Author Rebuttal · Reviewer_ZmMJ · 2026-04-03
> >
> > Thanks to the authors for the rebuttal and clarifications of some of my concerns. I do, however, have some further concerns.
> > - The authors pointed out that bounding $\min_t\sum_{t = 0}^{T-1} \eta_t \lVert \nabla J(V_t) \rVert^2$ by a constant "is a standard criteria for establishing convergence rate for nonconvex optimization".
> >    - The authors rightfully admit this does not generally imply last-iterate convergence and will fix the wording of the paper to reflect this. Then, given that last-iterate convergence is not guaranteed (nor parameter or loss convergence), my question is, **what kind of convergence is this**? How does bounding $\min_t\sum_{t = 0}^{T-1} \eta_t \lVert \nabla J(V_t) \rVert^2$ **imply** that gradient descent with LoRA converges to a stationary point? A quick view of (Theorem 2 of Jiang et al., 2024) and (Theorem 4.2 of Park & Klabjan, 2025) does not even answer these questions (I note that the first reference is published work, the second seems to be a pre-print).
> >    - I just want to add that bounding the running average $\frac{1}{T} \sum_{t = 0}^{T-1} \lVert \nabla \mathcal{J}(V_t) \rVert^2$ could imply convergence to a stationary point. I only provide an informal "intuitive" explanation . First, note that the terms of the sum are non-negative. Thus, if the average is close to zero, then every term of the sum must also be close to zero. Since the average is closer to zero the larger $T$ is, then we can see that  $\lVert \nabla \mathcal{J}(V_t) \rVert$ gets closer to zero as $t$ grows, i.e., convergence to a stationary point (or some group of them) happens. Note that this line of reasoning cannot be applied when upper bounding $\min_t\sum_{t = 0}^{T-1} \eta_t \lVert \nabla J(V_t) \rVert^2$.
> > - Thanks to the authors for pointing out why $V=0$ is a stationary point. However, I want to point out that lines lines 312-315 say that "LoRA **may** converge to the origin even if the original global minima is arbitrarily far away, explaining the observed divergence in behavior between LoRA and full-rank fine-tuning". First of all, the use of "may" is not formal and confuses the reader about whether the paper claims that convergence happens. Second, the authors say in the rebuttal that $V=0$ "**is not** necessarily attractive nor are the iterates guaranteed to converge to it". Then,
> >     - Why use the word "**may**" then? Wouldn't be more appropriate to eliminate the claim in lines 312-315?
> >     - If truly one can't say whether the origin $V=0$ is attractive or not, then what is the whole point of the theoretical analysis? The theoretical analysis should prove convergence, but the authors are telling me that convergence to the origin is not guaranteed.
> >     - It is possible that there exist *multiple* stationary points --- and such points may not be isolated, though my guess is that the origin *is* isolated. Can the authors check whether it is possible to prove that the origin is **locally** stable using their current results? Maybe linearization around the origin is a viable option?

---

> > > ### Author Response · Authors · 2026-04-04
> > >
> > > **On convergence to a stationary point.** Before we proceed, we assume that the reviewer made a typo in combining the $\min$ with the summation in their response. To clarify, our work shows that $\min_{t } \lVert \nabla J(V_t) \rVert^2$ converges to zero, which follows from bounding $\sum_{t = 0}^{T-1} \eta_t \lVert \nabla J(V_t) \rVert^2$ by a constant (and appropriate machinery regarding $\eta_t$).
> > >
> > >  In our original rebuttal, we focus on the standards in LoRA-adjacent work, including preprints, to address the reviewer's concern about "unfair comparison" to the existing literature. Now we would like to provide a clearer picture of these guarantees and how they relate to the broader nonconvex optimization literature as a whole.
> > >
> > > First, we clarify that convergence of $\frac{1}{T} \sum_{t = 0}^{T-1} \lVert \nabla J(V_t) \rVert^2$ does not necessarily imply last-iterate convergence, which we abbreviate as LIC. For example, some bounds in the literature are of the form
> > >
> > >  A. $\frac{1}{T} \sum_{t = 0}^{T-1} \lVert \nabla J(V_t) \rVert^2 \leq \frac{C_1}{\sqrt{T}}$
> > >
> > >  B. $\frac{1}{T} \sum_{t = 0}^{T-1} \lVert \nabla J(V_t) \rVert^2 \leq \frac{C_1}{T} + C_2$
> > >
> > > where $C_1$, $C_2$ are nonzero constants. These provide bounds on $\sum_{t = 0}^{T-1} \lVert V_t \rVert^2$ that *grow* over time and **do not imply LIC**. Works like (Theorem 5.5 of Chen et al., ICML 2025) and (Theorem 4.8 of Ghiasvand et al. 2025) employ these bounds to derive convergence rates.
> > >
> > > However, if we achieve an $O(1/T)$ bound of the form
> > >
> > > C. $\frac{1}{T} \sum_{t = 0}^{T-1} \lVert \nabla J(V_t) \rVert^2 \leq \frac{C}{T}$
> > >
> > > then we have $\lim_{T \to \infty} \sum_{t = 0}^{T-1} \lVert \nabla J(V_t) \rVert^2 < \infty$  and this does imply LIC.
> > >
> > > The reviewer asks: if LIC is not guaranteed, then what kind of convergence is this? Our response is simply, it is the convergence of the term $\min_t \lVert V_t \rVert^2$, which implies that the gradient norm vanishes at some point during the trajectory as $T \to \infty$, i.e. that we visit an approximate stationary point. We now point to some foundational texts that explain why this type of convergence is important and meaningful.
> > >
> > > In Section 1.2.3 of the textbook *Introductory Lectures on Convex Optimization*, Nesterov 2004, the author analyzes the convergence of GD, including the LIC derived from a bound like C. However, on page 28, the author states that to establish a *convergence rate,* we must bound the minimum gradient norm $g*_N$ over all $N$ iterates (1.2.15) to capture the rate of convergence of $g^*_N$ as $N \to \infty$. The author notes that we cannot say anything about the convergence **rate** of other values, such as the iterates or the function values.
> > >
> > > In Theorem 4.8, 4.9, and 4.10 of "Optimization Methods for Large-Scale Machine Learning," Bouttou, Curtis, and Nocedal 2018, the authors analyze the convergence rate of SGD. Thm 4.8 is a bound like B which *does not* imply LIC. Thm 4.9 demonstrates that the $\lim \inf$ of the gradient norm converges to zero, with authors observing that "*A “lim inf” result of this type should be familiar to those knowledgeable of the nonlinear optimization literature. After all, such a result is all that can be shown for certain important methods,*" and that the "*intuition*" from such a bound is that the "*gradient norms cannot stay bounded away from zero.*" Finally Thm 4.10 establishes a bound of the form $\lim_{T \to \infty} \sum_{t = 0}^{T - 1} \eta_t \lVert \nabla J(V_t) \rVert^2 < \infty$, which is also what we prove. In fact, the authors make it clear that LIC for SGD is only possible under much stronger requirements, in Corollary 4.12.
> > >
> > > The above literature establishes that **bounding the minimum gradient norm is an essential target to compare convergence rates of algorithms on nonconvex functions, in settings where LIC is not always guaranteed.** Our work fits into this context by deriving a convergence rate of LoRA GD, which compare with the rates established in other works for LoRA and LoRA-adjacent algorithms.
> > >
> > > **On $V = 0$ as a stationary point.** The main focus of this work is **not** showing that LoRA converges to the origin. We do not make this claim, as it is well known that gradient descent on a general nonconvex function may converge to any stationary point. In lines 312-315 (which is after we already establish the scope, context, prior literature, and main contributions of our paper), we state that LoRA *may* (as in, possible but not guaranteed) converge to the stationary point at $0$, providing a useful pathological example of how LoRA might differ in behavior from full-rank finetuning. This is logically consistent with stating that it "does not necessarily" occur. For maximum clarity, we will amend this sentence to "Consequently, it is possible (but not guaranteed) for LoRA to converge to the origin even if the original global minima is arbitrarily far away, an example of how LoRA and full-rank fine-tuning might lead to different outcomes."

---

### Official Review · Reviewer_rgti · 2026-03-09

**Soundness:** 3
**Presentation:** 3
**Significance:** 3
**Originality:** 2
**Overall Recommendation:** 4
**Confidence:** 3

**Summary:**

This paper addresses the gap in the theoretical understanding of the convergence behavior of the LoRA gradient descent algorithm. The authors provide the first non-asymptotic convergence rate analysis of the original LoRA algorithm, showing it converges to a stationary point at $O(1/\log T)$ under standard assumptions of smoothness and boundedness on the original loss - without requiring artificial bounds on the adapter matrices. The analysis is based on reformulating LoRA in terms of stacked adapter matrices and a “Lipschitz-like” descent lemma. Experimental results on logistic regression and ResNet-18 for CIFAR-10 empirically validate the theoretical rate and propose learning rate derived from the convergence theory that stabilize and accelerate LoRA training.

**Compliance With Llm Reviewing Policy:**

Affirmed.

**Final Justification:**

After reviewing process, I decide to maintain my score.

**Key Questions For Authors:**

1. Could the authors demonstrate that the condition for sublinear convergence holds in all the presented experiments?

2. If the boundedness condition on $|V_t |$ in Theorem 3.5 may not hold, I would like to see possible experiments confirming the absence of boundedness of this norm. From all the literature on LoRA fine-tuning, I am aware of fairly stable classical sublinear convergence across all application domains.

3. Why did the authors not consider the stochastic case, which is used in practice? What is the difficulty of obtaining theoretical estimates in this setting?

4. Could the authors hypothesize under which assumptions on the loss function sublinear convergence would hold?

5. Does the initialization of adapters affect the convergence guarantees?

6. Have you explored the sensitivity of convergence rates and stability to the choice of the scaling parameter $\alpha$ in your adaptive/normalized learning rate schemes? Are there regimes where these schedules underperform compared to simpler baselines?

**Limitations:**

yes

**Strengths And Weaknesses:**

**Strengths**

1. The learning rate selection strategies are not only derived from theory but also shown empirically to outperform baseline techniques (Figures 1 and 2 for logistic regression and ResNet-18).

2. The result explains an observed difference between LoRA and classic gradient descent analysis, filling a major theoretical gap.

3. Hyperparameters and codebase are provided, and key implementation details (e.g., use of learning rate finder, batchnorm deactivation) can be used directly.

**Weaknesses**

1. In all experiments, the algorithm exhibits standard sublinear convergence. I would like to see practical evidence on any task that the convergence can be characterized as $\mathcal{O}(1/ \log T)$.

2. Only CIFAR-10 is used for all experiments, and tasks/models are limited to logistic regression and ResNet-18. The major applied domain for LoRA - LLM fine-tuning remains untouched. This constrains generality.

3. The “position dependency” phenomenon is highlighted but only lightly explored empirically. Further ablations or deeper study would strengthen the narrative.

---

> ### Author Rebuttal · Authors · 2026-03-28
>
> We thank you for your positive remarks and constructive feedback. Based on your review, we ran several additional experiments and added the resulting figures to the front page of the anonymous GitHub repository, viewable at https://anonymous.4open.science/r/lora-C709/README.md. The figures and resulting insights will be added to the camera-ready version of the paper. Due to space constraints, we group some of the weaknesses and questions together in our responses.
>
> **(W1) (W3) (Q1) (Q2) (Q4)**
>
> We ran additional experiments comparing the convergence of (mini-batch) gradient descent with and without LoRA. The results in Figure A and B on logistic regression and ResNet-18 support that the convergence of LoRA gradient descent is slower.
>
> We note that the $O(1/\log T)$ rate derived in our work is a worst-case bound for general nonconvex functions that hinges on the potential growth of $\lVert V_t \rVert^2$ over time. This has no effect on standard gradient descent, but can slow down LoRA gradient descent. If the parameters remain within a bounded space, the convergence rate is $O(1/T)$, but this is not always guaranteed in practice.
>
> The experimental settings in our paper showcase these two potential outcomes. For the ResNet-18 model, the iterates converge to a global minima and $\lVert V_t \rVert$ stops growing after a finite number of iterates. However, for the logistic regression model, $\lVert V_t \rVert$ consistently grows over 60 epochs. We ran additional experiments extending the logistic regression training to 1000 epochs. Figure C demonstrates that although the loss changes very little, both $\lVert V_t \rVert$ and  $\lVert V_t \rVert^2$ continue to grow unbounded as $t$ increases. These results on the logistic regression model (which most closely aligns with our theoretical setting) suggests that the convergence is indeed meaningfully slower than $O(1/T)$.
>
> **(W2) (W3) (Q5)**
>
> Finally, we scaled our experiments to the LLM setting. We fine-tuned TinyLlama-1.1B-Chat-v1.0 on the Alpaca dataset using LoRA with a rank of 32. Figure D demonstrates that if the LLM is initialized with small parameters, the adaptive learning rates maintain faster and more stable convergence than constant learning rates of similar magnitudes. However, Figure E demonstrates that if the LLM is initialized with large values, the advantage of the learning rate schedules is diminished. This is  because in this very high-dimensional space, $\lVert V_t\rVert^2$ is not just very large but also grows extremely slowly relative to its size. As a result, as shown on the right side of Figure E, the adaptive learning rate looks similar to a constant learning rate. This suggests that in this regime, the convergence may appear to the naked eye as $O(1/T)$ for some finite number of iterations, although the algorithm can still asymptotically converge at the rate $1/\log T$ as $T \to \infty$. We will add this important context to our paper.
>
> **(Q3)**
>
> We agree that the stochastic case would be a natural next step of this work. A major challenge of this setting is identifying appropriate noise assumptions. In the standard analysis, a typical assumption is that the variance/second-moment of the noise is uniformly or relatively bounded. This second-moment bound is then leveraged in the classic descent lemma
>
> $J(V_{t+1}) \leq J(V_t) + \langle \nabla J(V_t), V_{t+1} - V_t \rangle + \lVert V_{t+1} - V_t \rVert^2 $
>
> to handle the squared term $\lVert V_{t+1} - V_t \rVert^2$.
>
> However, in our work the descent lemma (Lemma 3.3) contains several higher order terms with powers of 3 and 4. Therefore, the convergence depends on the third or fourth moment of the noise, which are not necessarily bounded even if the second moment is bounded. One approach for addressing this is to simply assume that all moments are bounded, but this is not very interesting. Another approach is to develop a reasonable assumption on the fourth moment based on the finite-sum structure of typical empirical risk minimization problems. Balancing a realistic assumption with the requirements for convergence is the primary difficulty of this setting.
>
> **(Q6)**
>
> Much like a constant learning rate, if $\alpha$ is too small or too large, the algorithm will underperform by either converging too slowly or displaying unstable behavior respectively. However, we found that the behavior of the adaptive/normalized learning rates are generally less sensitive than the constant learning rates, which might display very fast convergence for some large $\eta$ but become unstable for any values above that. As shown in Figure 1, the adaptive/normalized learning rates can take advantage of a wide range of values while maintaining numerical stability. Finally, as discussed above, in the regime of high dimensional parameters and large norms, the learning rate schedules behave more closely like constant learning rates.

---

> > ### Author Rebuttal · Reviewer_rgti · 2026-04-03
> >
> > Thank you to the reviewers for the detailed response. Having read all the reviews, I decide to maintain my score.

---

### Official Review · Reviewer_2Wdc · 2026-03-09

**Soundness:** 2
**Presentation:** 3
**Significance:** 3
**Originality:** 3
**Overall Recommendation:** 4
**Confidence:** 3

**Summary:**

This paper studies the convergence of LoRA algorithm. The authors prove that under the Lipschitz smoothness of the original loss function, LoRA has a log convergence rate. Under additional bounded norm conditions, it has polynomial convergence match standard gradient descent. Numerical results are provided to validate the theoretical findings.

**Compliance With Llm Reviewing Policy:**

Affirmed.

**Final Justification:**

Thanks for the rebuttal. I will increase the score.

**Key Questions For Authors:**

See the weakness part above.

**Limitations:**

yes

**Strengths And Weaknesses:**

## Strength

1. The convergence of LoRA is a long-standing problems. This paper gives a neat solution without additional assumptions.

2. The writing is clear and easy to follow. The authors provide intuitive as well as technical explanations to help the readers understand the content.

## Weaknesses

1. The log convergence of LoRA is relatively slow. Do you think it is tight and can not be improved? Does experiments support that LoRA is far slower than standard gradient descent?

2. I have some questions on the correctness of the proof. For example, where does the equality come from on the line of 665? There should be a \|V_1\| in the third term. If this is an actual error, the descent lemma will need a major change.

---

> ### Author Rebuttal · Authors · 2026-03-28
>
> We thank you for your positive remarks and constructive feedback. In the following, we address the concerns raised in your review.
>
> **The log convergence of LoRA is relatively slow. Do you think it is tight and can not be improved? Does experiments support that LoRA is far slower than standard gradient descent?**
>
> We ran additional experiments comparing (mini-batch) gradient descent with and without LoRA for the same constant learning rates, and added the resulting figures to the front page of the anonymous GitHub repository, viewable at https://anonymous.4open.science/r/lora-C709/README.md. The results in Figure A and B on logistic regression and ResNet-18 support that the convergence of LoRA gradient descent is slower.
>
> To explore whether the bound may be tight, we examined more closely the empirical behavior of $\lVert V_t \rVert$ and $\lVert V_t \rVert^2$. The reason for this is that the $1/\log T$ slowdown stems from the growth of  $\lVert V_t \rVert^2$ which is upper bounded by a term that is linear in $t$. Since the learning rate $\eta_t$ is inversely proportional to $\lVert V_t \rVert^2$, this yields the harmonic series $\sum_{t = 0}^{T-1} \eta_t$, which grows as $\log T$. For the ResNet-18 model, it appears that $\lVert V_t \rVert$ stops growing after a finite number of iterates. However, for the logistic regression model, $\lVert V_t \rVert$ exhibits a consistent upward trend over 60 epochs. We extended the logistic regression training to 1000 epochs, and we added the figures to the anonymous GitHub repository. Figure C plots both $\lVert V_t \rVert$ and  $\lVert V_t \rVert^2$, demonstrating that although the loss does not change much, the parameter norm continues to grow monotonically as $t$ increases. The results on the logistic regression model (which most closely aligns with our theoretical setting) suggests that the bound could indeed be close to tight in settings where the iterates converge to a stationary point at infinity, such as if the loss function decreases to zero at infinity. However, empirically $\lVert V_t \rVert^2$ appears to be growing slightly less than linearly, suggesting room for improvement. .
>
> **I have some questions on the correctness of the proof. For example, where does the equality come from on the line of 665? There should be a |V_1| in the third term. If this is an actual error, the descent lemma will need a major change.**
>
> Thank you for pointing this out. We indeed made an error and dropped a factor of $\lVert V_t \rVert$ in line 665. Fortunately, after carefully combing through the rest of the proof, we find that this error has a minimal impact, and the correction actually simplifies the descent lemma by turning the $ \lVert V_2 - V_1 \rVert^3 $ term into $ \lVert V_2 - V_1 \rVert^3 \lVert V_1 \rVert$, allowing it to be combined with the other $ \lVert V_2 - V_1 \rVert^3 \lVert V_1 \rVert$ term. In the response to Reviewer 1BxX, we provide the corrected proof in detail, starting at line 665.
>
> To summarize, Lemma 3.3, the descent lemma, is now
>
> $\mathcal{J}(V_2) \leq \mathcal{J}(V_1) + \langle \nabla \mathcal{J}(V_1), V_2 - V_1 \rangle_F +  \sqrt{2} L\lVert V_2 - V_1 \rVert^2  \lVert V_1 \rVert^2  + \sqrt{2} L \lVert V_2 - V_1 \rVert ^3 \lVert V_1 \rVert + \frac{\sqrt{2} L }{4}  \lVert V_2 - V_1\rVert^4  + \lVert \nabla \mathcal{L} (E_1 V_1 V_1^T E_2 ) \rVert \lVert V_2 - V_1 \rVert^2 . $
>
> Because there is one fewer term, the later proofs are also simplified, and we obtain that the descent in one step (9) can be improved by a constant factor from $5$ to $4$, to $\mathcal{J}(V_{t+1}) \leq \mathcal{J}(V_t) - \frac{\eta_t}{4}$, with the step size requirement (9) also loosened by a constant to $\eta_t = \min \{\frac{1}{4 \sqrt{2} L(\lVert V_{t} \rVert^2 + \lVert \nabla \mathcal{L}(E_1 V_{t} V_{t}^T E_2) \rVert) }, 1 \}$. Since the impact is just a change in a constant, the rest of the proof logic follows from this point almost identically. We have made these necessary changes in the manuscript and they will appear in the camera-ready version.

---

> > ### Author Rebuttal · Reviewer_2Wdc · 2026-04-04
> >
> > Thanks for the rebuttal, I will increase the score.

---

### Official Review · Reviewer_1BxX · 2026-03-10

**Soundness:** 4
**Presentation:** 4
**Significance:** 4
**Originality:** 4
**Overall Recommendation:** 6
**Confidence:** 5

**Summary:**

This paper derives the convergence rate of LoRA GD on a learning rate policy given by the L-smooth level of loss function. It makes a good and clear theoretical contribution by relaxing the L-smoothness and bounded property of both LoRA blocks A and B, to be only requiring L-smoothness of their product layer block parameter W=BA.
This paper provide concrete theoretical analysis and experiments to prove that, under given learning rate schedule, LoRA gradient descent algorithm can converge to statinnary point at rate O(1/logT).

**Compliance With Llm Reviewing Policy:**

Affirmed.

**Final Justification:**

With detailed justification and correction on the proof details, this paper made a fundamental theoretical contribution on LoRA convergence.

It neatly remove the L-smooth condition of both A,B block and only inherit the layer W to be L-smooth (standard in NN convergence analysis), without any problematic additional assumption added.

As in maths, the neater assumption, the harder to prove. While the paper prove the whole part step by step very clear and smooth to read (much unlike today's LLM generated proofs that is abused in ML papers which contains much jump-of-steps and strange claims without justification), and the conclusion is satisfactory:

O(1/logT) and O(1/T) for bounded model parameters. Bounded condition is common as layernorm/clipping is a common component in today's LLM, the O(1/T) can be a good convergence rate in practice and O(1/logT)  be a slow rate guarantee for arbitrary NN.

The paper finally also provide extension to multiple matrices, brooding the scope from single layer to whole model (as whole model parameters is stack of different layer parameters).


It is a solid theoretical paper with neat assumption and beautiful proof that is rare and valuable in today's ML research environment, I highly recommend an acceptance by AC.

**Key Questions For Authors:**

1, the learning rate (8) has a term norm(V_t)^2 in the denominator. When for large models the term can grows very large, incurring a slow learning rate and slow convergence. Could you give more insights into how the convergence is with model size?

2, your derivation in general is for reparameterization W = g(V), where in LoRA g(V) = E1VVTE2. In your derivation you use some properties of g such as lemma A.2. As LoRA has more than W=BA form for some variations (for example, in SVD form such as in AdaLoRA). I am wondering what is the necessary property of g so that your original derivation will make sense. So that your work can be more general. This is optional and you may not answer in the rebuttal period if find it too hard.

3, I am concerned about Reviewer 2Wdc's second weakness. As it is a factual error in maths derivation and the following derivations are based on it. If the author need more space to write the correct version of full revised proof, you can neglect my questions 1 and 2 as they are optional (and far less important than the factual fault), so that you can use full space to show the revised proof as it is the most important part need to be checked.

**Limitations:**

yes

**Strengths And Weaknesses:**

Soundness:
theoretical:

I have checked all the derivations line by line in the appendix without detecting any error (except for typos).

this paper gives all the theorems and lemmas clearly stated and with rigorous proof. The assumptions is more general compared with previous work.

Experimental: this work only performs on small models and datasets, which is a limitation but not crucial as the main contribution is in theory.

Presentation: The theoretical proof and experiment results are well posted. This paper provide assumptions, decent lemma, and convergence rate on learning rate on a standard convergence analysis format. The appendix is well structured and the proofs are concrete without jumping steps, it is pleasant and smooth to read the proofs (except for one typo in line 772, missing a sqrt(2) before L in the second term).


)

Significance: This paper derive LoRA convergence while removing two key assumptions in previous LoRA convergence works: upper bound on LoRA blocks and L-smoothness for both LoRA blocks, which is a general and significant contribution.

Originality: This paper derive LoRA convergence on a new assumption condition, much more general than previous works, and provide a new learning rate scheduling policy.

---

> ### Author Rebuttal · Authors · 2026-03-28
>
> We sincerely thank you for your review and the opportunity to clearly state the proof correction in detail. We indeed made an error and dropped a factor of $\lVert V_t \rVert$ in line 665. Fortunately, we find that this error has a minimal impact, and the correction actually simplifies the descent lemma by turning the $ \lVert V_2 - V_1 \rVert^3 $ term into $ \lVert V_2 - V_1 \rVert^3 \lVert V_1 \rVert$, allowing it to be combined with the other $ \lVert V_2 - V_1 \rVert^3 \lVert V_1 \rVert$ term. In the following, we provide the corrected proof starting at line 665. We highlight in red changes to the expressions.
>
> $\lVert \nabla \mathcal{J}(V_1 + td) - \nabla \mathcal{J}(V_1) \rVert  \leq \sqrt{2} Lt \lVert d \rVert   \lVert V_1 + td\rVert^2 + \sqrt{2} Lt \lVert d \rVert \lVert V_1 \rVert^2 + \textcolor{red}{\sqrt{2} L t^2 \lVert d \rVert^2 \lVert V_1 \rVert}  + 2 t \lVert \nabla \mathcal{L} (E_1 V_1V_1^T E_2) \rVert \lVert d\rVert$
>
> $\int_0^1 \lVert \nabla \mathcal{J}(V_1 + td) - \nabla \mathcal{J}(V_1) \rVert \lVert d \rVert dt \leq  \sqrt{2} L \lVert d \rVert ^2 \int_0^1 t \lVert V_1 + td \rVert^2 dt + \sqrt{2} L \lVert d \rVert^2  \lVert V_1 \rVert^2 \int_0^1 t dt + \textcolor{red}{ \sqrt{2} L \lVert d \rVert^3 \lVert V_1 \rVert \int_0^1 t^2 dt} + 2 \lVert \nabla \mathcal{L} (E_1 V_1 V_1^T E_2 ) \rVert \lVert d \rVert^2 \int_0^1 t dt$
>
> $= \sqrt{2} L \lVert d \rVert ^2( \frac{1}{2} \lVert V_1 \rVert^2 + \frac{2}{3} \langle V_1,  d\rangle_F + \frac{1}{4} \lVert d \rVert^2)+ \frac{\sqrt{2}}{2} L\lVert d \rVert^2  \lVert V_1 \rVert^2  + \textcolor{red}{L \frac{\sqrt{2}}{3} \lVert d \rVert^3 \lVert V_1 \rVert} + \lVert \nabla \mathcal{L} (E_1 V_1 V_1^T E_2 ) \rVert \lVert d \rVert^2 $
>
> $\leq \frac{\sqrt{2}}{2} L \lVert d \rVert ^2 \lVert V_1 \rVert^2 + \frac{2\sqrt{2}}{3} L \lVert d \rVert ^3 \lVert V_1 \rVert + \frac{\sqrt{2}}{4} L \lVert d \rVert ^4 + \frac{\sqrt{2}}{2} L\lVert d \rVert^2  \lVert V_1 \rVert^2  + \textcolor{red}{L \frac{\sqrt{2}}{3} \lVert d \rVert^3 \lVert V_1 \rVert}  + \lVert \nabla \mathcal{L} (E_1 V_1 V_1^T E_2 ) \rVert \lVert d \rVert^2 $
>
> $= \textcolor{red}{\sqrt{2} L \lVert d \rVert ^3 \lVert V_1 \rVert} + \frac{\sqrt{2}}{4} L \lVert d \rVert^4 +  \sqrt{2} L\lVert d \rVert^2  \lVert V_1 \rVert^2  +  \lVert \nabla \mathcal{L} (E_1 V_1 V_1^T E_2 ) \rVert \lVert d \rVert^2.$
>
> So the descent lemma is now
> $\mathcal{J}(V_2) \leq \mathcal{J}(V_1) + \langle \nabla \mathcal{J}(V_1), V_2 - V_1 \rangle_F + \textcolor{red}{\sqrt{2} L \lVert V_2 - V_1 \rVert ^3 \lVert V_1 \rVert} + \sqrt{2} L\lVert V_2 - V_1 \rVert^2  \lVert V_1 \rVert^2   + \frac{\sqrt{2}}{4} L \lVert V_2 - V_1\rVert^4  + \lVert \nabla \mathcal{L} (E_1 V_1 V_1^T E_2 ) \rVert \lVert V_2 - V_1 \rVert^2.$
>
> Applying the descent lemma with $V_1 = V_t$, $V_2 = V_{t+1} = V_t - \eta_t \nabla \mathcal{J}(V_t)$ yields
> $\mathcal{J}(V_{t+1}) \leq \mathcal{J}(V_t) - \eta_t \lVert \nabla \mathcal{J}(V_t) \rVert^2 + \textcolor{red}{\sqrt{2} L \eta_t^3 \lVert \nabla \mathcal{J}(V_t) \rVert ^3 \lVert V_t \rVert} +  \eta_t^2\lVert \nabla \mathcal{J}(V_t)\rVert^2 ( \sqrt{2} L\lVert V_t \rVert^2 + \lVert \nabla \mathcal{L} (E_1 V_t V_t^T E_2 ) \rVert ) + \frac{\sqrt{2} L }{4} \eta_t^4 \lVert \nabla \mathcal{J}(V_t)\rVert^4.$
>
> As before, we pick $\eta_t$ to minimize the last three terms to guarantee descent in function value. Since there is one less term, we change the constant $5$ to $4$ to keep the proof clean (although it is not strictly necessary). Therefore, we choose
> $\eta_t = \min \\{\frac{1}{\textcolor{red}{4} \sqrt{2} L(\lVert V_{t} \rVert^2 + \lVert \nabla \mathcal{L}(E_1 V_{t} V_{t}^T E_2) \rVert) }, 1 \\}$. Following the same logic established in our paper, we obtain the same bounds on $\eta_t$ up to constant factors:
>
> $\eta_t \leq \frac{1}{\textcolor{red}{4} \sqrt{2} L\lVert V_{t} \rVert^2 + \textcolor{red}{4}\lVert \nabla \mathcal{L}(E_1 V_{t} V_{t}^T E_2) \rVert} \leq \left( \frac{1}{\textcolor{red}{64} \sqrt{2} L \lVert V_t \rVert^2 \lVert  \nabla \mathcal{L}(E_1 V_{t} V_{t}^T E_2) \rVert }\right)^{1/2}$
> $\leq \left( \frac{\textcolor{red}{1}}{ \textcolor{red}{4} \sqrt{2} L \lVert \nabla \mathcal{J}(V_t) \rVert \lVert V_t \rVert } \right)^{1/2},$
>
> and
>  $\eta_t  \leq \left( \frac{\textcolor{red}{1}}{\textcolor{red}{27} \sqrt{2} L \lVert \nabla \mathcal{J} (V_t) \rVert^2 }\right)^{1/3} \leq \left( \frac{1}{ \sqrt{2} L \lVert \nabla \mathcal{J} (V_t) \rVert^2 }\right)^{1/3}.$
>
> Because we were able to eliminate the $\lVert \nabla \mathcal{J}(V_t) \rVert^3$ term, we no longer require lines 749-796. This leads to the improved function descent in one step lemma (Lemma 3.4):
>
> $\mathcal{J}(V_{t+1})  \leq \mathcal{J}(V_{t}) - \frac{\eta_t}{\textcolor{red}{4}} \lVert \nabla \mathcal{J}(V_{t}) \rVert^2.$
>
> The rest of the proof logic follows from this point almost identically, since all major expressions are the same except for changing the constant 5 to 4. We have made these changes and they will appear in the final camera-ready version.

---

> > ### Author Rebuttal · Reviewer_1BxX · 2026-04-02
> >
> > The response is correct.
> >
> > I figured it out myself with a shorter version of correction:
> > As (21) has a very loose inequality (1/5 multiple from (20)), and considering the missed additional V_1 only raise the coefficient 2/3 to 1, (3/2multiple), as 1/5 * 3/2 = 3/10 <1, the proof need no major correction. (just change the loose inequality of 1/5 multiple to be 3/10 multiple, other no change and line 749-796 no need)
> >
> > I read other reviews' comments and think that most are from misunderstanding or inclarification, without any hard weakness.
> >
> > I would raise my score to original.

---

### Decision · Program_Chairs · 2026-04-30

**Decision:**

Accept (spotlight)

**Comment:**

All reviewers agree that the paper is technically strong, well-written, and addresses a timely and important problem related to the theoretical understanding of LoRA-based optimization. They highlight the novelty of the analysis, the strength and clarity of the theoretical results, and the relevance of the problem to modern large-scale learning. The paper is also praised for its clean presentation and for providing insights that could be of broad interest to the community.

Any minor concerns raised by reviewers were adequately addressed in the rebuttal and do not affect the overall positive assessment. Given the strong reviewer consensus, solid technical contributions, and clear relevance, I recommend acceptance.